# Polyketide synthase-derived sphingolipids mediate microbiota protection against a bacterial pathogen in *C. elegans*

Lena Peters [1], Moritz Drechsler[2,3], Michael A. Herrera[4], Jing Liu[2], Barbara Pees[1], Johanna Jarstorff[1], Anna Czerwinski [1], Francesca Lubbock[4], Georgia Angelidou [5], Liesa Salzer[6], Karlis Arturs Moors [7], Nicole Paczia [5], Yi-Ming Shi [2,8], Hinrich Schulenburg [1,9], Christoph Kaleta [7], Michael Witting[10,11], Manuel Liebeke [12,13], Dominic J. Campopiano [4,17] ✉, Helge B. Bode[2,3,14,15,16,17] ✉ & Katja Dierking [1,17] ✉

Protection against pathogens is a major function of the gut microbiota. Although bacterial natural products have emerged as crucial components of host-microbiota interactions, their exact role in microbiota-mediated protection is largely unexplored. We addressed this knowledge gap with the nematode *Caenorhabditis elegans* and its microbiota isolate *Pseudomonas fluorescens* MYb115 that is known to protect against *Bacillus thuringiensis* (Bt) infection. We find that MYb115-mediated protection depends on sphingolipids (SLs) that are derived from an iterative type I polyketide synthase (PKS) cluster *Pf*SgaAB, thereby revealing a non-canonical pathway for the production of bacterial SLs as secondary metabolites. SL production is common in eukaryotes but was thought to be limited to a few bacterial phyla that encode the serine palmitoyltransferase (SPT) enzyme, which catalyses the initial step in SL synthesis. We demonstrate that *Pf*SgaB encodes a pyridoxal 5'-phosphate-dependent alpha-oxoamine synthase with SPT activity, and find homologous putative PKS clusters present across host-associated bacteria that are so far unknown SL producers. Moreover, we provide evidence that MYb115-derived SLs affect *C. elegans* defence against Bt infection by altering SL metabolism in the nematode host. This work establishes SLs as structural outputs of bacterial PKS and highlights the role of microbiota-derived SLs in host protection against pathogens.

A major function of the gut microbiota is its contribution to host protection against pathogens[1]. The protective mechanisms conferred by the gut microbiota are complex and include direct competitive or antagonistic microbe–microbe interactions and indirect microbe-host interactions, which are mediated by the stimulation of the host immune response, promotion of mucus production, and maintenance of epithelial barrier integrity[2]. Microbiota-derived metabolites are known to play an important role in the crosstalk between the gut microbiota and the immune system[3–5]. Of these metabolites, bacterial natural products have emerged as crucial components of host-microbiota interactions[6–8].

Bacterial natural products (also called secondary or specialised metabolites) are chemically distinct, often bioactive compounds that

are not required for viability, but mediate microbial and environmental interactions[9]. Some of the most studied natural products include polyketides, which are derived from polyketide synthase (PKS). PKS are found in many bacteria, fungi, and plants, and produce structurally diverse compounds by using an assembly line mechanism similar to fatty acid synthases[10]. Many PKS-derived natural products show potent antibiotic (e.g., erythromycin and tetracycline), antifungal (e.g., amphotericin and griseofulvin) or immunosuppressant (e.g., rapamycin) activities[11] and have thus long played a central role in advancing therapeutic treatments for a wide range of medical conditions. The majority of characterised polyketides were isolated from free-living microbes, while only a few are known to be gut microbiota-derived[8]. Most well-studied examples of PKS-derived products from the microbiota are virulence factors associated with pathogenicity[12]. Few PKS-encoded natural products were reported to play a role in microbiota-mediated protection against pathogens both directly and indirectly. For example, the antifungal polyketide lagriamide supports direct symbiont-mediated defence of eggs against fungal infection in the beetle *Lagria villosa*[13]. A PKS cluster of the rodent gut symbiont *Limosilactobacillus reuteri* is required for activating the mammalian aryl hydrocarbon receptor, which is involved in mucosal immunity[14]. Additionally, *L. reuteri* PKS was recently demonstrated to exhibit antimicrobial activity and to drive intraspecies antagonism[15]. Yet, the vast majority of microbiota-encoded PKS are of unknown function and mechanistic studies linking specific microbial natural products to host phenotypes are scarce.

The *Pseudomonas fluorescens* isolate MYb115 belongs to the natural gut microbiota of the model organism *Caenorhabditis elegans*[16]. It was previously found that MYb115 protects *C. elegans* against the harmful effects of infection with *Bacillus thuringiensis* (Bt) without directly inhibiting pathogen growth, likely through an indirect, host-dependent mechanism[17,18]. The nature of the microbiota-derived protective molecule and the involved host processes were unknown. Here, we identify a biosynthetic gene cluster (BGC) in MYb115 encoding an iterative type I PKS that is required for MYb115-mediated protection and produces sphingolipids (SLs). Thus, we discovered a non-canonical pathway for the production of bacterial SLs, which relies on a BGC, the *P. fluorescens* PKS cluster *Pf*SgaAB. Hence SLs are produced as secondary metabolites. We additionally demonstrate that MYb115-derived SLs affect *C. elegans* SL metabolism and establish the importance of *C. elegans* SL metabolism for survival after Bt infection.

## Results

### *P. fluorescens* MYb115 PKS cluster is required for *C. elegans* protection against Bt infection

The natural microbiota isolate *P. fluorescens* MYb115 protects *C. elegans* against infection with the Gram-positive pathogenic *B. thuringiensis* strain Bt247 likely through a host-dependent mechanism[18], but the nature of the microbiota-derived protective molecule was unknown. We performed an antiSMASH analysis[19] of the MYb115 genome to identify natural product BGCs. We found three BGCs in the MYb115 genome, encoding a non-ribosomal peptide synthetase (NRPS), an iterative type I PKS cluster, and an arylpolyene pathway.

We modified the PKS and NRPS clusters of MYb115 by inserting the inducible arabinose $P_{BAD}$ promoter. Thus, while induction of BGC expression requires arabinose supplementation, no expression should be observed in the absence of arabinose supplementation, mimicking a deletion phenotype[20]. We assessed the ability of MYb115 $P_{BAD}sga$ (MYb115 PKS cluster, Fig. 1A) and MYb115 $P_{BAD}nrpA$ (MYb115 NRPS cluster, Fig. 1B) in an induced (+ arabinose) and non-induced (− arabinose) state to protect *C. elegans* against Bt247 infection. We found that infected *C. elegans* exposed to induced MYb115 $P_{BAD}sga$ showed significantly increased survival when compared to infected worms on MYb115 $P_{BAD}sga$ in a non-induced state (Fig. 1A, C and Supplementary Data 1). Arabinose supplementation had no effect on resistance of *C.*

*elegans* to Bt infection on its standard laboratory food *Escherichia coli* OP50 (Supplementary Data 1). While the PKS gene cluster affects MYb115-mediated protection, we did not observe significant differences in worm survival with or without arabinose supplementation on the MYb115 $P_{BAD}nrpA$ strain (Fig. 1B and Supplementary Data 1). We therefore focused on the *P. fluorescens* MYb115 PKS gene cluster (hereafter *Pf*SgaAB) in our subsequent analyses.

We then deleted either, the PKS SgaA (MYb115 Δ*sgaA*), or the alpha-oxoamine synthase (AOS) SgaB (MYb115 Δ*sgaB*) (Fig. 1D) to confirm the requirement of *Pf*SgaAB in MYb115-mediated protection. While MYb115 provided significant protection against infection in *C. elegans* compared to worms on *E. coli* OP50 (Fig. 1E[18]), protection of worms on both MYb115 mutants was lost (Fig. 1E). Protection in MYb115 Δ*sgaA* was restored upon expression of SgaAB from the vanillic acid-inducible $P_{vanCC}$ promoter on a plasmid (pSEVA631) (Fig. 1F). These results clearly demonstrate a role of *Pf*SgaAB in MYb115-mediated protection.

### *P. fluorescens* MYb115 PKS cluster (*Pf*SgaAB) produces long chain sphinganines and phosphoglycerol SLs

MYb115-mediated protection against Bt247 infection depends on the PKS cluster *Pf*SgaAB. We next asked which natural product is produced by *Pf*SgaAB. Using LC-MS, we identified three compounds that are produced in MYb115 $P_{BAD}sga$ upon induction with arabinose (Fig. 1C). We subsequently established that the compounds **1**–**3** are also produced, but less abundant, in MYb115, and that both MYb115 deletion mutants (MYb115 Δ*sgaA*, and MYb115 Δ*sgaB*) are not able to produce compounds **1**–**3** (Fig. 1G). MS[2] experiments revealed that compounds **1**–**3** show structural similarities to commercially available long chain sphinganines (Fig. S1). We determined the molecular composition through isotopic labelling experiments and confirmed that compounds **1**–**3** are very long chain sphinganines (C24, C26, C28, Fig. S2A–C).

Moreover, we performed lipidomic analysis of MYb115 using high-resolution Liquid Chromatography Tandem Mass Spectrometry (HRES-LC-MS/MS) and found that in addition to the three sphinganines **1**–**3** MYb115 produces compounds **4**–**6**, each with masses 154 Da heavier than those of the three sphinganine derivatives (Fig. S2D–F and Supplementary Data 3). Since the masses of **4**–**6** did not match any known lipids in the MS-DIAL LipidBlast (version 68) dataset, we used the exact mass and different lipid headgroups to propose structures for compounds **4**–**6**. We conclude that compounds **4**–**6** are most likely phosphoglycerol sphingolipids (PG-sphingolipids). Next, we analysed the relative abundance of sphinganines **1**–**3**, and PG-sphingolipids **4**–**6** in MYb115 and MYb115 $P_{BAD}sga$ induced by arabinose or repressed by glucose supplementation. While the sphinganines **1**–**3** were more abundant in the induced MYb115 $P_{BAD}sga$ samples, the total abundance of PG-sphingolipids **4** and **5** did not differ compared to MYb115 supplemented with arabinose (Fig. S2G). Thus, increase in sphinganine production does not necessarily lead to increase in PG-sphingolipid production.

Many BGCs are silent under typical laboratory conditions and activation of secondary metabolic pathways can be a challenge[21]. Indeed, we observed substantial variations in the protective effect of MYb115 under our standard laboratory conditions over the course of this project. We hypothesized that the variations in the protective effect are related to variations in SL production. To test this hypothesis, we used MYb115 $P_{BAD}sga$ and MYb115 Δ*sgaA*/$P_{vanCC}sgaAB$, in which SL production can be activated by arabinose and vanillic acid supplementation, respectively, and that produce SLs at much higher levels than wildtype MYb115 (Fig. S3). We harvested MYb115 $P_{BAD}sga$ and MYb115 Δ*sgaA*/$P_{vanCC}sgaAB$ pellets of the same bacterial cultures, whose protective effect we then tested in survival analyses, and visualized SL production using MALDI mass spectrometry spot assays[22]. SL production indeed varied between different bacterial cultures, even under conditions of targeted induction. Most importantly,

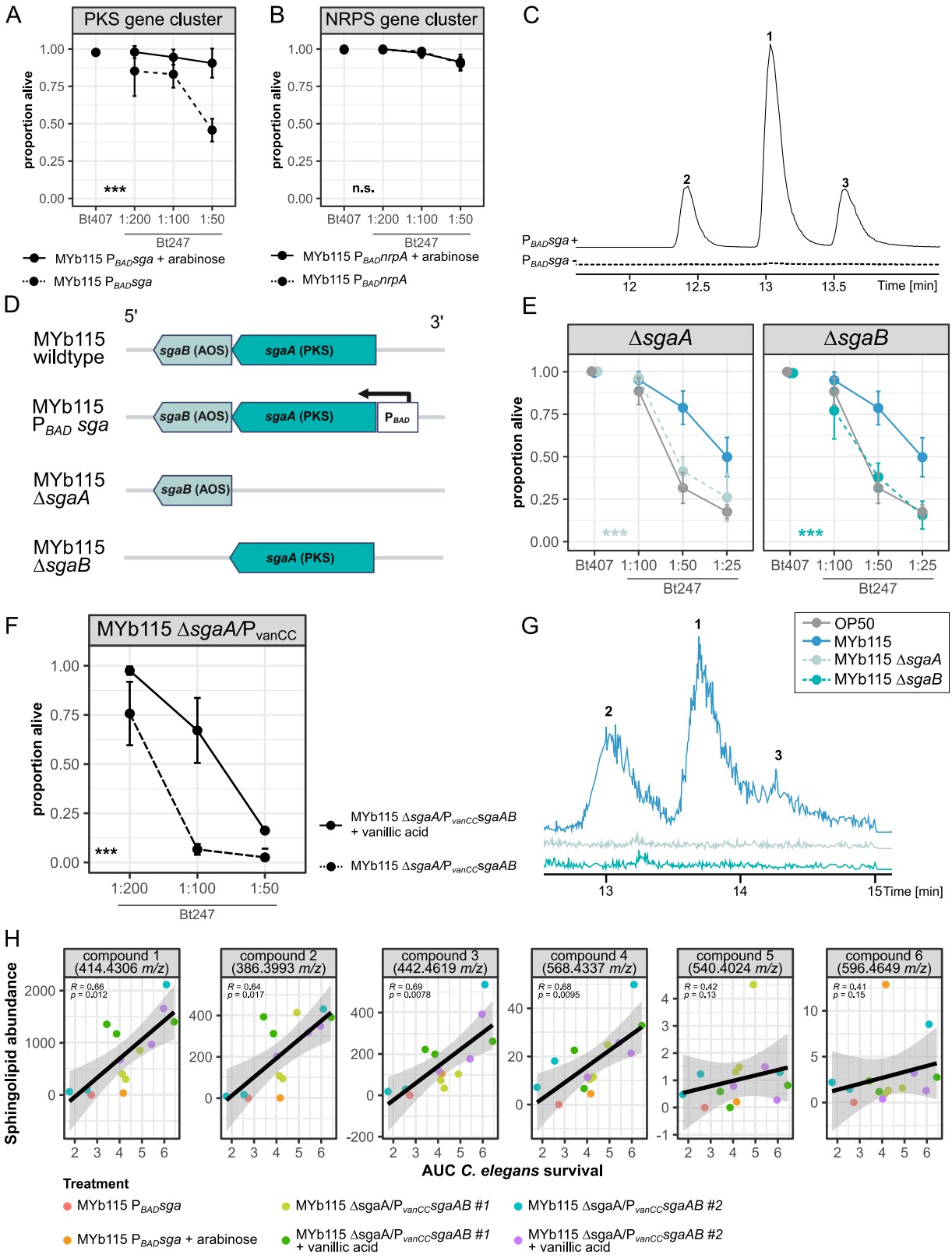

we found that abundances of sphinganines **1**–**3** and PG-sphingolipid **4** correlate significantly with worm survival following Bt247 infection (Fig. 1H and Supplementary Data 1), providing further evidence that host protection is dependent on these SLs. Since our results also demonstrate that SL production and the associated protective effect is variable under the given laboratory conditions, we always controlled for the protective effect in our experiments.

## A proposed pathway for iT1PKS cluster-dependent SL biosynthesis

SL synthesis in bacteria and eukaryotes involves the condensation of an amino acid (typically serine in mammals) and a fatty acid (typically palmitate in mammals) via the serine palmitoyltransferase (SPT) enzyme that uses pyridoxal phosphate (PLP) as cofactor for serine decarboxylation and coupling to palmitoyl-CoA[23]. In the case of MYb115, the

**Fig. 1 | MYb115 PKS cluster-derived SLs mediate protection against *B. thuringiensis* infection. A, B** Survival proportion of *C. elegans* N2 on *P. fluorescens* MYb115 P$_{BAD}$*sga* (**A**) or MYb115 P$_{BAD}$*nrpA* (**B**) induced with arabinose (solid line) or in a non-induced state without arabinose supplementation (dashed line) 24 h post infection with *B. thuringiensis* Bt247. Bt407 was used as a non-pathogenic control. The data shown is representative of three independent runs with four replicates each (see Supplementary Data 1). **C** LC-MS chromatogram of MYb115 P$_{BAD}$*sga* extracts from cultures with (solid line) and without (dashed line) arabinose supplementation. Upon induction with arabinose, three compounds (**1–3**) are produced. **D** Schematic representation of the MYb115 PKS gene cluster and its modifications. Polyketide synthase (PKS) SgaA, alpha-oxoamine synthase (AOS) SgaB and inducible arabinose promoter (P$_{BAD}$). **E** Survival proportion of N2 on *E. coli* OP50, MYb115, or MYb115 knockout mutants. *C. elegans* on both tested mutants MYb115 Δ*sgaA*, and MYb115 Δ*sgaB* were significantly more susceptible ($p = 1.17E-09$ or $p = 2.00E-16$, respectively) to infection with Bt247 than worms on wildtype MYb115. Means ± standard deviation (SD) of $n = 4$, are shown in survival assays

(**A, B, E**), $n = 3$ in (**F**). Statistical analyses were carried out using the generalized linear model (GLM) framework with a binomial distribution. All tests were two-sided, and *p*-values were adjusted for multiple comparisons using the Bonferroni correction. Significance is indicated as ****p* < 0.001. **F** Survival proportion of N2 on MYb115 Δ*sgaA*/P$_{vanCC}$*sgaAB*, which expresses SgaAB under the vanillic acid-inducible P$_{vanCC}$ promoter on the pSEVA631 plasmid. Survival was assessed 24 h post-infection with Bt247, comparing vanillic acid-induced (solid line) and non-induced (dashed line) conditions ($p = 3.53E-11$). **G** LC-MS chromatogram of MYb115 wt, Δ*sgaA* and Δ*sgaB*. **H** Correlation of area under the *C. elegans* survival curve (AUC) and peak intensity, representing bacterial SL abundance. Each facet represents the correlation for a specific bacterial SL compound (**1–6**), with different bacterial treatments indicated by colour. Correlations were calculated using the two-sided Spearman method, correlation coefficients are shown with 95% confidence intervals. Source data and additional survival runs are provided in Supplementary Data 1.

protective SLs **1–3** and PG-sphingolipid **4**, are produced by the two-gene cluster *PfsgaAB* (**S**phinga**n**ine biosynthesis **A** and **B**), in which *sgaA* encodes a PKS and *sgaB* encodes an AOS with predicted structural homology to SPTs deposited on the Protein Data Bank (https://www.rcsb.org/ PDB: 2JG2; 2X8U; 3A2B; 8GUH, Fig. S4). Like its SPT homologues, *Pf*SgaB is expected to condense fatty acyl-thioesters with L-serine to give 3-ketodihydrosphinganine (3-KDS)-like intermediates, which is the first committed step in SL biosynthesis (Fig. 2A[24,25]). To confirm this activity, the gene encoding *Pf*SgaB was codon-optimised, synthesised and cloned into pET28a with an N-terminal poly-His tag for downstream purification (Figs. S5 and S6). Recombinant expression in *E. coli* BL21(DE3) resulted in yellow-tinged biomass, from which *Pf*SgaB was purified to homogeneity using tandem cobalt-IMAC and size-exclusion chromatography (SEC, Fig. S7). SEC analysis provided an estimated molecular weight (MW) of 108 kDA, consistent with protein dimerization (Fig. S8). We first incubated recombinant, purified *Pf*SgaB with varying concentration of L-serine and observed PLP:L-serine external aldimine formation by UV–vis spectroscopy (413 nm), with and estimated K$_d$ = 1.30 ± 0.0256 mM (Figs. 2B and S9). Following this, we probed *Pf*SgaB condensation activity using acyl-CoAs **7–9** as a surrogate co-substrates (Fig. 2C), capturing the CoASH by-product using 5,5′-dithio-bis(2-nitrobenzoic acid) (DTNB, Fig. S10A). Using this colorimetric assay, a clear response was obtained when *Pf*SgaB was incubated with both L-serine and **7**, in contrast to all negative controls (Fig. S10B). Furthermore, *Pf*SgaB shows virtually exclusive preference for L-serine over deoxysphingolipid-forming amino acids L-alanine and glycine (Fig. 2D). Similar condensation activity with L-serine was also observed when **8** and **9** were used as co-substrates (Fig. S10C). We subsequently identified all corresponding 3-KDS products **10–12** by LC/ESI-MS ($m/z$ = 300.29, 328.33, 356.37, see Fig. 2E, F). Moreover, through isotopic labelling experiments we could show that $^{13}C^{15}N$-labelled serine is incorporated during sphinganine biosynthesis in MYb115 in vivo (Supplementary Data 2). Taken all together, this data confirms the functional assignment of *Pf*SgaB as a SPT and the key gateway into SL biosynthesis in MYb115.

Furthermore, we identified a putative short chain dehydrogenase/reductase (SDR) in the MYb115 SL BGC (locus ID: KW062_RS19775), which shares homology with several eukaryotic 3-ketodihydrosphinganine reductases (KDSR, see Figs. S11 and 12). KDSR homologues are also found in fungal BGCs that produce SL-like, PKS-derived mycotoxins such as fumonisin (FUM13, Uniprot: W7LL82)[26] and sphingofungin (SphF, Uniprot: B0XZV2)[27]. KDSR catalyses the reduction of 3-KDS to dihydrosphinganine (DHS)[28,29]; whilst this step is ubiquitous in eukaryotic SL biosynthesis, it is unusual in bacterial SL pathways outside of *Bacteroides* and *Prevotella* species[30,31]. Taken together with gene context, we propose that this enzyme, hereafter named *Pf*SgaC, mediates 3-KDS reduction in MYb115. The inclusion of this eukaryotic-like step further distinguishes this BGC from canonical bacterial SL biosynthesis.

## Homologous PKS clusters are present across diverse bacterial genera

Iterative PKS were originally found in fungi and only rarely in bacteria[10]. However, a large number of bacterial iterative PKS were identified more recently[32]. While only a few bacterial iterative PKS and their products have been studied, our work is to our knowledge the first example of a PKS cluster shown to be involved in SL biosynthesis and also the first description of a *P. fluorescens* isolate as SL producer. We explored the distribution of the two-gene *PfsgaAB* in bacteria listed in the NR NCBI database and found 6,101 homologous putative PKS clusters (Supplementary Data 4). Interestingly, the homologous PKS clusters were present in bacteria that are known to be closely associated with hosts, including human pathogens and opportunistic pathogens (Fig. 3A). When we analysed the distribution of the target BGC class at the genus level, we found that the putative PKS cluster is dominantly distributed in *Burkholderia* (Fig. 3B). Interestingly, *Burkholderia pseudomallei* K96243 has previously been shown to produce sphingosine-1-phosphate lyases[33,34], but like *Pseudomonas*, *Burkholderia* is not yet a known SL producer. The fact that we found the potential PKS cluster SgaAB in *Burkholderia* suggests that they may be able to produce sphingosine-1-phosphate and not just degrade it.

## MYb115-derived SLs modulate the expression of genes related to pathogen defence and contribute to intestinal barrier protection

In a first step towards exploring the function of microbiota-derived SLs in mediating the interaction with the host, we tested whether MYb115-produced SLs affect the ability of MYb115 to colonise the host or modulate host feeding behaviour. We did not observe a difference in host colonisation between MYb115 and MYb115 Δ*sgaA* (Fig. S13A and Supplementary Data 5), nor did we see differences in *C. elegans* feeding behaviour on MYb115 and MYb115 Δ*sgaA* (Fig. S13B and Supplementary Data 5).

MYb115 protects *C. elegans* against Bt infection without directly inhibiting pathogen growth, likely through an indirect, host-dependent mechanism[18]. When grown on MYb115 *C. elegans* is also protected against infection with another Bt strain, Bt679, that produces distinct pore-forming toxins (PFTs) (Fig. S14[17,18]) and this protection also depends on bacterial SL production (Fig. S14 and Supplementary Data 6). We thus considered that activation of general host defence mechanisms may contribute to MYb115-mediated protection and performed gene expression profiling of 1-day adult worms on either protective MYb115 or non-protective MYb115 Δ*sgaA* in the absence and presence of pathogenic Bt247 (Fig. 4A and Supplementary Data 7). We did not observe any genes differentially regulated between worms on SL-producing MYb115 and worms on the MYb115 Δ*sgaA* mutant when using an adjusted *p*-value cutoff of 0.05. This may indicate that MYb115-derived SLs do not strongly affect *C. elegans* on the transcript level, but

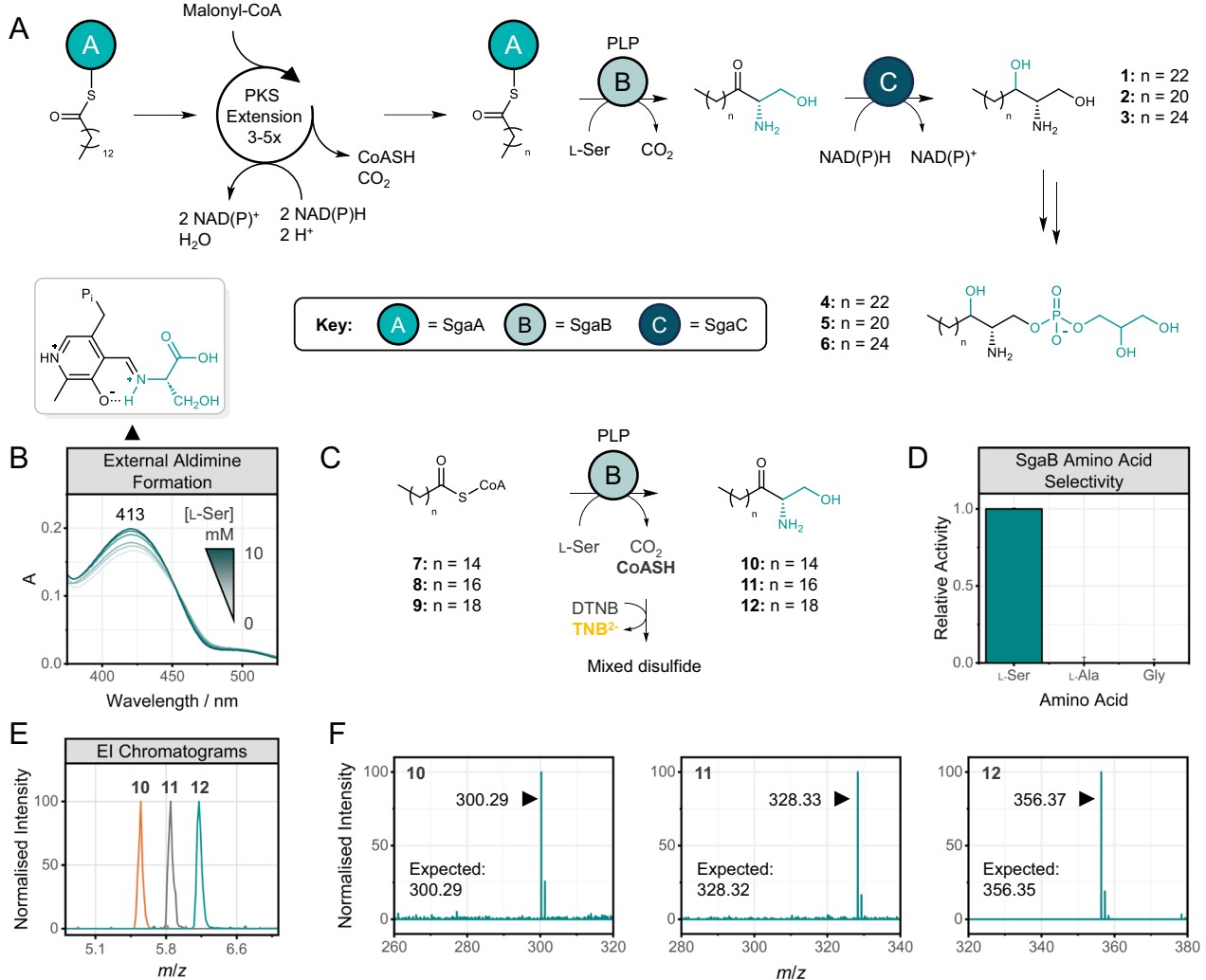

**Fig. 2 | Proposed biosynthesis of *P. fluorescens* MYb115 PKS cluster *Pf*SgaAB-derived SLs. A** Biosynthesis scheme of MYb115-derived PG-sphingolipids **4–6**. The production of 3-ketodihydrosphinganines (KDSs) is catalysed by the iterative PKS (iPKS) *Pf*SgaA and PLP-dependent serine palmitoyltransferase (SPT) *Pf*SgaB. The reduction of KDSs to dihydrosphinganines **1–3** is presumably catalysed by the KDS reductase homologue *Pf*SgaC. **B** PLP external aldimine formation following the addition of up to 10 mM L-serine (L-ser), monitored by UV–vis spectroscopy. External aldimine formation is signified by an increase in absorbance at 413 nm. **C** Schematic representation of *Pf*SgaB-catalysed decarboxylative condensation between acyl-CoAs **7–9** and L-ser to give 3-ketodihydrosphinganines **10–12**. **D** Relative activity of *Pf*SgaB in the presence of $C_{16}$-CoA **7** and L-serine, L-alanine or glycine, determined using the DTNB assay (412 nm). UV–vis measurements were recorded after 20 min of incubation. Error bars represent the standard deviation of three technical replicates. All measurements were corrected for non-specific background absorbance. **E** Extracted ion (EI) chromatograms of *Pf*SgaB-derived products **10–12**, detected by LC/ESI-MS. **F** [M + H]⁺ ions of *Pf*SgaB-derived products **10–12**, detected by LC/ESI-MS. The theoretical *m/z* is shown for each product.

more strongly influence the host on the proteome or metabolome level. Using a less stringent cutoff (non-adjusted *p*-value < 0.01), we nevertheless identified 122 differentially expressed (DE) genes between the two treatments in the absence of Bt247 (23 genes were down regulated and 99 genes upregulated in worms on MYb115 Δ*sgaA* (Supplementary Data 7) and 48 DE genes in the presence of Bt247 (22 genes were down regulated and 26 genes upregulated in worms on MYb115 Δ*sgaA* (Supplementary Data 7). Genes that are related to *C. elegans* pathogen defence were indeed enriched in both gene sets, including targets of known pathogen defence pathways, such as the p38 and JNK-like MAPK pathways (Fig. 4B, C and Supplementary Data 7). However, we did not find any evidence of the involvement of the p38 MAPK and the JNK MAPK KGB-1 in decreasing MYb115-mediated protection against Bt247 (Fig. 4B, C and Supplementary Data 8).

The damage caused by Bt PFTs leads to loss of intestinal barrier function and we have previously shown that MYb115 limits Bt-induced damage to the intestinal epithelium[18]. Here, we used a *C. elegans* strain expressing PGP-1::GFP, a labelled ATP binding-cassette transporter,

whose expression is restricted to the apical plasma membrane of the intestinal epithelium[35], to test if MYb115-derived SLs are involved in mitigating Bt-induced damage. Indeed, in Bt247 infected worms on arabinose-induced MYb115 P*BAD*sga, we found a clear reduction of the relocalisation of the PGP-1::GFP marker to intracellular vesicles (Fig. 5 and Supplementary Data 9), which is regarded as a response to membrane damage caused by PFTs[36]. In contrast, there was no difference in the numbers of intracellular vesicles between infected worms on *E. coli* OP50 and the MYb115 Δ*sgaA* mutant (Fig. 5), suggesting that the SLs produced by MYb115 P*BAD*sga contribute to protection of the intestinal barrier following Bt infection.

## MYb115-derived SLs alter host fatty acid and SL metabolism

Mouse lipid metabolism was previously shown to be affected by gut microbiota-derived SLs[37]. Moreover, in a *C. elegans* Parkinson disease model, the probiotic *B. subtilis* strain PXN21 protects the host against protein aggregation by modulating SL metabolism[38]. Thus, we hypothesised that MYb115-derived SLs impact host metabolism. To

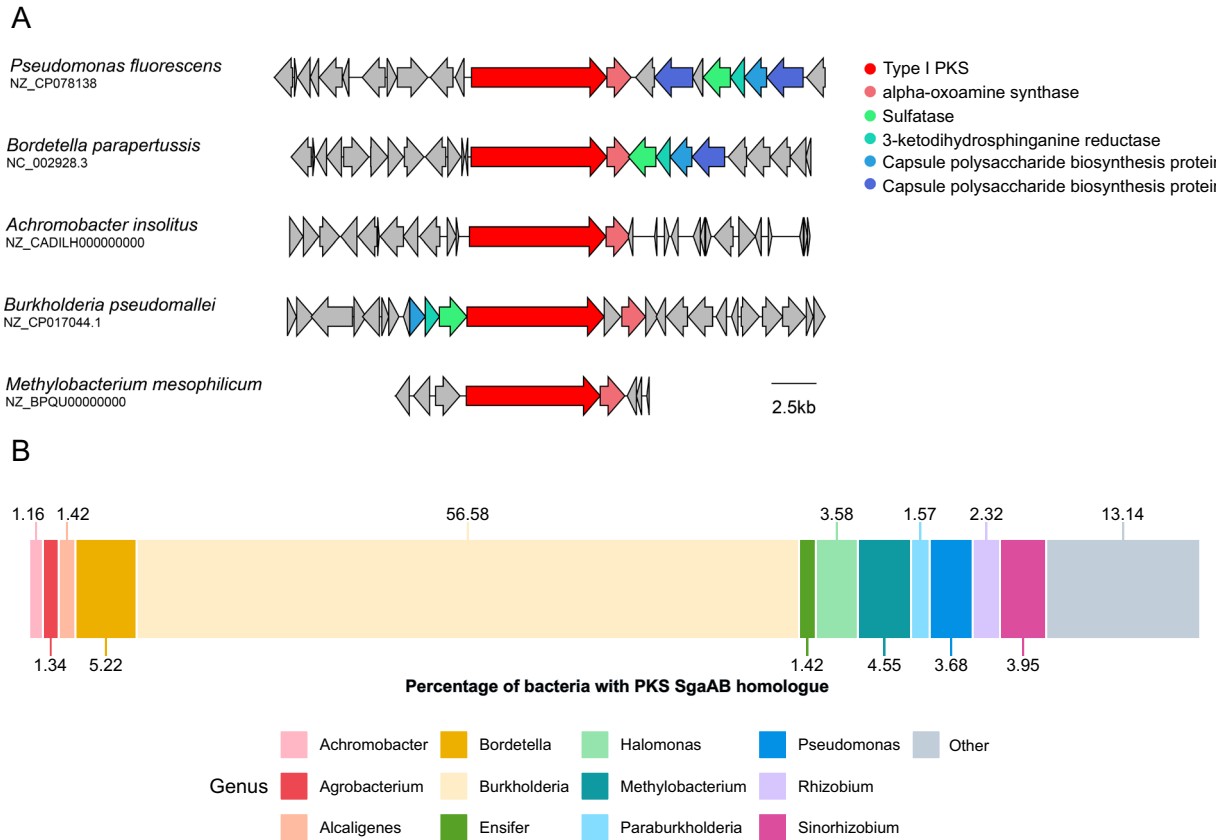

**Fig. 3 | Distribution of *P. fluorescens* MYb115 PKS cluster SgaAB homologues in bacteria.** The monomodular PKS (KW062_RS19805) and the alpha-oxoamine synthase (KW062_RS19800) in *P. fluorescens* MYb115 (NZ_CP078138) were searched against the NR NCBI database (https://www.ncbi.nlm.nih.gov/) using cblaster (1.8.1)[89]. **A** Five representative PKS cluster SgaAB homologs from various bacterial genera aligned and visualised using clinker[90]. **B** Total distribution of 6101 PKS cluster SgaAB homologs across different bacterial genera. The width of each box represents the percentage of all identified PKS cluster SgaAB homologs, found in each bacterial genus are provided as source data in Supplementary Data 4.

test this hypothesis, we integrated the transcriptomic data into the iCEL1314 genome-scale metabolic model of *C. elegans*[39] to create context-specific models, simulating metabolite flow through the *C. elegans* reactions network under specific treatment conditions (see methods). Statistical analysis of the reaction fluxes resulted in 16 (Bt247 +) and 16 (Bt247 −) significant differences when comparing MYb115 Δ*sgaA* and MYb115 worms (Supplementary Data 10). Through a pathway enrichment analysis of the significant reactions against a background of our model pathways, we found that in the absence of Bt247, animals colonised by MYb115 or MYb115 Δ*sgaA* varied in the activity of multiple pathways linked with SL precursor production, such as monomethyl branched chain fatty acid biosynthesis, as well as SL metabolism itself (Fig. 6A and Supplementary Data 10). In the presence of Bt247, propanoate metabolism was most strongly affected (Fig. S15A and Supplementary Data 10). Under both infection and non-infection conditions, the valine, leucine, and isoleucine degradation pathway were significantly enriched. This pathway degrades branched-chain amino acids and is directly connected with propanoate metabolism that provides components for the synthesis of the C15iso fatty acid, which is the precursor for SLs in *C. elegans*[40]. We also focused on SL metabolism directly: Flux variability analysis[41] revealed a significant difference in upper bound values for the SL metabolism reactions in worms infected with Bt247 on MYb115 *versus* MYb115 Δ*sgaA* (*t*-test *p*-value < 0.001). Among those reactions, six reactions that all have ceramide as a substrate or product had the strongest changes (Fig. S15B and Supplementary Data 10). Overall, these findings suggest that worms colonised by MYb115 *versus* MYb115 Δ*sgaA* have a significantly reduced capacity to generate SLs.

## MYb115-derived SLs interfere with *C. elegans* complex SLs

The metabolic network analysis revealed that SL metabolism reactions show differential activity between MYb115 and MYb115 Δ*sgaA*. To confirm that MYb115-derived SLs affect *C. elegans* SL metabolism, we performed lipidomic profiling of *C. elegans* exposed to MYb115 or MYb115 Δ*sgaA*. We identified *C. elegans* SLs by manual interpretation of MS[1] and MS[2] data and used SLs that have previously been described in *C. elegans* containing a C17iso-branched chain sphingoid base and different length of N-Acyl chains as input[42] (Supplementary Data 11). Since the employed analytical method cannot separate between different hexoses attached to the SL, they were annotated as hexosylceramides (HexCers), which showed the neutral loss of 162.052275 Da. Monomethylated phosphoethanolamine glucosylceramides (mmPEGCs), a class of *C. elegans* phosphorylated glycosphingolipids, were identified based on fragments as previously described[43,44].

We were not able to detect MYb115-derived sphinganine in worms on MYb115. Likewise, we did not detect any SLs based on sphinganines produced by MYb115. A possible explanation is that bacterial sphinganine concentrations in worms are below the detection limit. However, we found different complex host SLs based on the C17iso-branchend chain sphingoid base typical for *C. elegans* with N-acyl sides of length 16–26 without or with hydroxylation. In addition to previously established SLs, we identified HexCer with an additional hydroxyl group instead of the double bond in the sphingoid base. In total, we identified 40 *C. elegans* SLs from different SL classes. We did not observe a difference in *C. elegans* C17iso sphinganine or C17iso sphingosine, but in certain dihydroceramide (DhCer) and ceramide

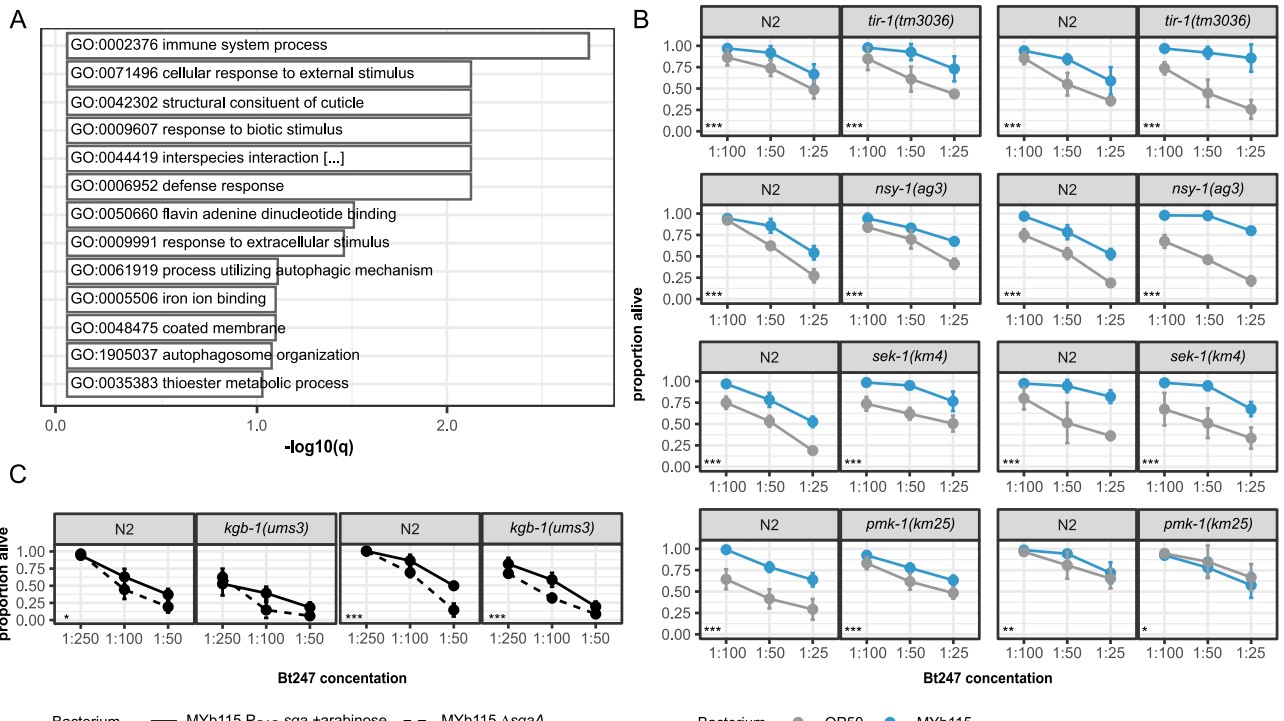

**Fig. 4 | MYb115-mediated protection is independent of known *C. elegans* pathogen defense pathways. A** Transcriptional response of *C. elegans* to MYb115-derived SLs. Enrichment analysis of genes differentially regulated between worms exposed to SL-producing MYb115 and worms exposed to non-SL producing MYb115 Δ*sgaA* in the presence of pathogenic Bt247 (Supplementary Data 7). **B**, **C** Survival of p38 and JNK MAPK pathway mutants. Means ± standard deviation (SD) of *n* = 4 (p38 MAPK pathway (**B**)); *n* = 3 (*kbg-1(ums3)* survival (**C**)) are shown in all survival assays. Statistical analyses were carried out using the generalized linear model (GLM)

framework with a binomial distribution. All tests were two-sided, and *p*-values were adjusted for multiple comparisons using the Bonferroni correction. Significance is indicated as, \*\*\**p* < 0.001, \*\**p* < 0.01, \**p* < 0.05. All *p*-values can be found in Supplementary Data 8. *nsy-1(ag3)* and *sek-1(km4)* share the same N2 control since the experiment was conducted in parallel, with statistical analysis adjusted accordingly, as highlighted in Supplementary Data 8. Source data are provided in Supplementary Data 8.

(Cer) species between worms on MYb115 or MYb115 Δ*sgaA*. Also, complex SLs downstream of ceramides, i.e., sphingomyelins (SMs) and HexCers were increased in worms on MYb115 Δ*sgaA*, and some even significantly increased (Fig. 6B). Individual SL profiles are shown in Fig. S16. Most of the significant changes occurred at the lower or upper end of the detected N-acyl chain length. No changes occurred in SLs containing an N-acyl of 22 or 24 carbon length. However, the series of SM(d33:1, d35:1, d37:1), showed a consistent and significant increase. Additionally, SM(t37:1) and SM(t43:1) as well as the corresponding HexCer(t37:1) and SM(t37:1) increased significantly. Notably, we found the highest fold-changes between MYb115 and MYb115 Δ*sgaA*-exposed worms for mmPEGC. However, changes were not significant and so far, the biosynthesis pathway of mmPEGCs is unknown.

Together, our data suggest that MYb115-derived SLs interfere with *C. elegans* SL metabolism mainly at the conversion of dihydroceramide and ceramide to sphingomyelins and hexosylceramides.

### Modifications in *C. elegans* SL metabolism affect defence against Bt247 infection

Since MYb115 affects host SL metabolism and protects the worm against Bt infection, we next asked whether alterations in nematode SL metabolism affect *C. elegans* survival following Bt infection. We performed survival experiments using several *C. elegans* mutants of SL metabolism enzymes (Figs. 7A–C and S17 and Supplementary Data 12). We assessed the general involvement of SL metabolism in the response to Bt infection in the presence of the non-protective lab food *E. coli* OP50. We found that mutants of the *C. elegans* serine palmitoyl transferases *sptl-1(ok1693)* and *sptl-3(ok1927)*, which catalyse the de novo synthesis of the C17iso sphingoid base, and the ceramide

synthase mutants *hyl-1(ok976)* and *hyl-2(ok1766)*) showed increased survival on Bt in comparison to wildtype N2 worms (Fig. 7C), whereas the survival phenotype of two ceramide metabolic gene mutants, namely *cgt-1(ok1045)* and *cerk-1(ok1252)*, was variable (Fig. 7C). *cgt-1* encodes one of three *C. elegans* ceramide glucosyltransferases that generate glucosylceramides (GlcCers). *cerk-1* is a predicted ceramide kinase that catalyses the phosphorylation of ceramide to form ceramide-1-phosphate (C1P). In contrast, the *sms-1(ok2399)* mutant was clearly more susceptible to Bt247 infection than wildtype worms. *sms-1* encodes a *C. elegans* sphingomyelin synthase that catalyses the synthesis of sphingomyelin from ceramide. Accordingly, the *asm-3(ok1744)* mutant, which lacks the enzyme that breaks down sphingomyelin to ceramide, showed increased resistance to Bt247 (Fig. 7C). Notably, the ceramidase mutants *asah-1(tm495)* and *asah-2(tm609)* were also significantly more susceptible to Bt247 infection than the *C. elegans* control (Fig. 7C). *asah-1* encodes a *C. elegans* acid ceramidase that converts ceramide to C17iso-sphingosine, which is subsequently phosphorylated by the sphingosine kinase SPHK-1 to C17iso-sphingosine-1-phosphate[45]. Together, these results suggest that inhibition of de novo synthesis of ceramide and inhibition of the conversion of ceramide to GlcCer or C1P increases survival of *C. elegans* infected with Bt247, while inhibition of the conversion of ceramide to sphingomyelin or sphingosine decreases survival of Bt247-infected animals.

To elucidate the role of specific SLs in defence against Bt infection we supplemented Bt infected worms with the commercially available SLs ceramide, sphingomyelin, and sphingosine-1-phosphate. We found that supplementation with C18 and C20 ceramide significantly improved survival rates, while C22 ceramide, C16 sphingomyelin,

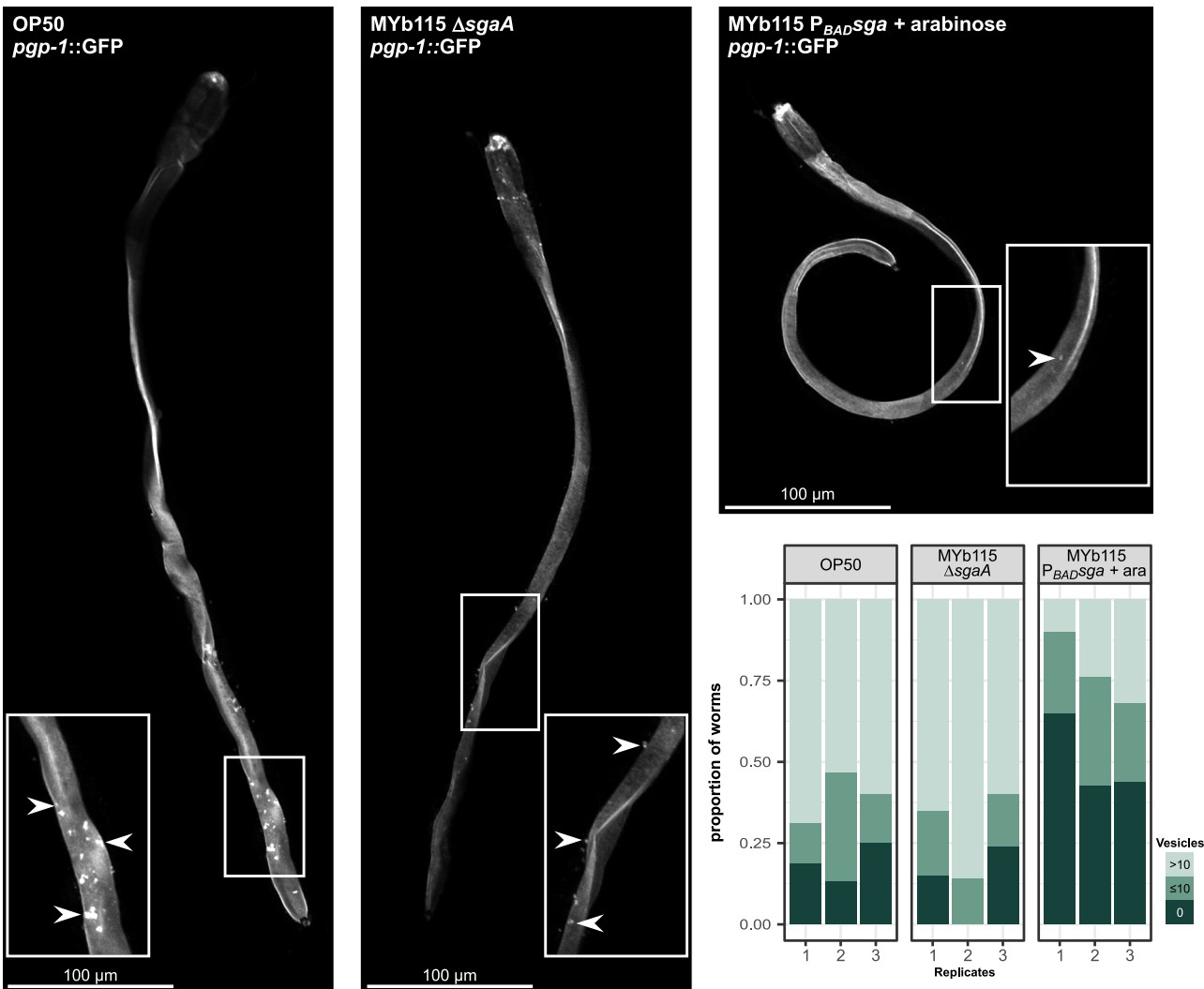

**Fig. 5 | MYb115-derived SL contribute to intestinal barrier protection.** Visualisation and quantification of vesicular structures following Bt247 infection. Worms were raised on either *E. coli* OP50, *P. fluorescens* MYb115 Δ*sgaA* or *P. fluorescens* MYb115 P$_{BAD}$*sga* + arabinose for 72 h and then infected with Bt247. Confocal images of PGP-1::GFP were captured 4 h after exposure to Bt247 mixed with either OP50, MYb115 Δ*sgaA* or MYb115 P$_{BAD}$*sga* + arabinose. For each worm all PGP-1::GFP positive vesicles were scored and categorised into either of the three groups "0 vesicles", "≤ 10 vesicles" or "> 10 vesicles". Representative images of worms are shown, highlighting magnified regions of PGP-1::GFP positive vesicles (indicated by white arrows) following Bt247 infection. Scale bar: 100 μm. The proportions of worms in each category are displayed as stacked bar plots for each replicate. Population size varied between 14 and 25 individuals (*n* = 3). Source data are provided in Supplementary Data 9.

C18 sphingomyelin, d-sphingosine, and S1P did not affect survival (Fig. S18 and Supplementary Data 13). These findings and the phenotypic differences between the ceramide metabolic gene mutants imply that the inhibition of ceramide metabolism (and the associated increase in ceramide content) is not the only factor determining susceptibility to infection.

We additionally assessed *C. elegans* SL metabolism mutant survival on the protective microbiota isolate MYb115. MYb115 and the inhibition of de novo synthesis of ceramide or the breakdown of sphingomyelin to ceramide protect worms against infection with Bt247. Therefore, we did not expect to see an effect of MYb115 on the increased survival phenotype of the *sptl-1, -3, hyl-1, -2,* and *asm-3* mutants. Our results are fully consistent with these expectations, the mutants were more resistant to Bt infection also on MYb115 (Fig. 7C). However, both ceramidase mutants *asah-1(tm495)* and *asah-2(tm609)*, which were more susceptible to Bt247 infection on *E. coli* OP50, were as susceptible as and even more resistant than wildtype worms on MYb115, respectively (Fig. 7C). Notably, MYb115 also ameliorated the susceptibility phenotype of the *sms-1(ok2399)*

mutant (Figs. 7C and S17). These data indicate that MYb115 interacts with host SL metabolism at least at the conversion of ceramide to sphingomyelin and C17iso-sphingosine.

## MYb115-mediated protection is independent of the *C. elegans* mitochondrial surveillance response and a Bt toxin glycosphingolipid receptor

Our data suggest a mechanistic link between microbiota-mediated alterations in host SL metabolism and protection against Bt infection. As a step towards exploring the underlying mechanisms, we tested two potential links between SLs and *C. elegans* defence against Bt infection. First, we explored the possible involvement of the mitochondrial surveillance response, which requires ceramide biosynthesis[46] and second we explored the role of complex glycosphingolipids that are receptors of the Bt Cry toxin Cry5B[47]. Bt247 infection did not induce the expression of the mitochondrial stress-induced *hsp-6*p::*gfp* reporter, neither did MYb115 (Fig. S19A), indicating that mitochondrial surveillance is not involved in MYb115-mediated protection against Bt247 infection. Also, the Bt-toxin

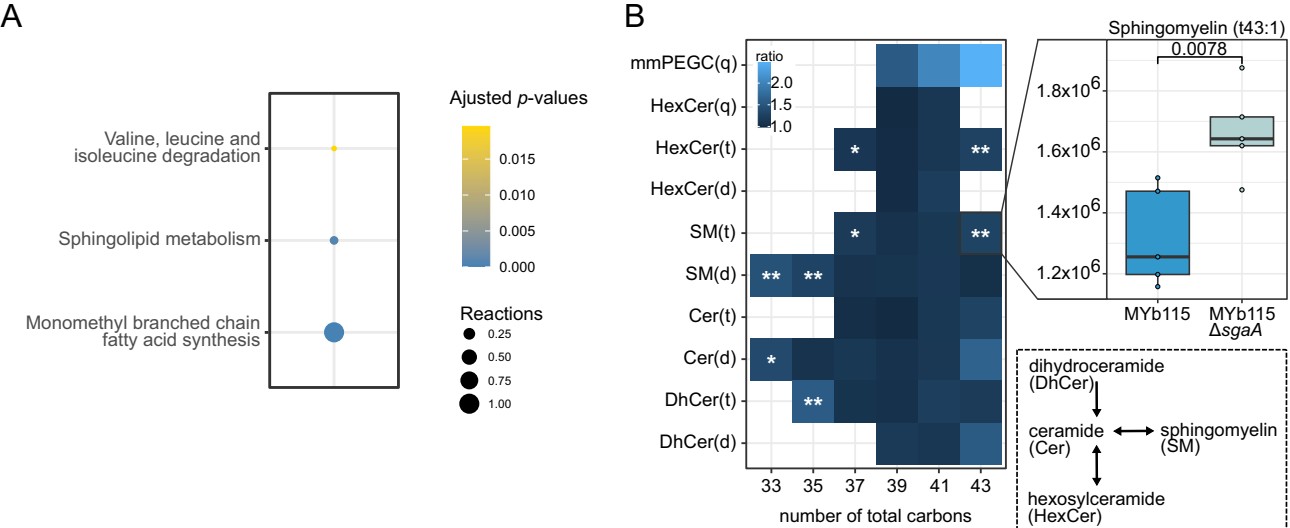

**Fig. 6 | MYb115-derived SLs modulate host SL metabolism. A** Enriched metabolic pathways if the *C. elegans* (iCEL1314) metabolic model were identified following a comparison of worm models integrated with transcriptome data from worms treated with MYb115 with worms treated with MYb115 Δ*sgaA*. Significant reactions obtained by calculating two-sided *p*-values from linear regression models (data ~ treatment) of FVA centres and OFD data layers were used for Flux Enrichment Analaysis (FEA) against the background of all reactions within the iCEL1314 *C. elegans* metabolic model. Benjamini-Hochberg was applied only for FEA output due to high pathway/reaction collinearity. Source data are provided in Supplementary Data 10. **B** Reduced SL contents in worms exposed to MYb115 compared to worms exposed to MYb115 Δ*sgaA*. The heatmap shows the differences in ratio of detected SLs between the mean of MYb115 Δ*sgaA* and the mean of MYb115. The boxplot shows the difference in ratio of Sphingomyelin (t43:1) in worms exposed to MYb115 Δ*sgaA* and MYb115, all remaining boxplots can be found in Fig. S16. Boxplots display the median (line), the first and third quartiles (box edges), and whiskers extending to the smallest and largest values within 1.5× the interquartile range. Points beyond this range are shown as outliers. Statistical analysis was done with a two-sided Welch's *t*- test ($n = 5$), * *p*-value < 0.05, ** *p*-value < 0.01. Dihydroceramides (DhCer), Ceramides (Cer), Sphingomyelins (SM), Hexosylceramides (HexCer), with hydroxylated fatty acyls (t) or non-hydroxylated fatty acyls (d), Hexosylceramides with phytosphingosine base and hydroxylated fatty acyls (HexCer(q)), monomethyl phosphoethanolamine glucosylceramide (mmPEGC(q)). Source data are provided in Supplementary Data 11.

resistant (bre) mutants *bre-2(ye31) and bre-3(ye26)*, which are defective in the biosynthesis of the Cry5B glycosphingolipid receptor[47], are susceptible to Bt247 infection and still protected by MYb115 (Fig. S19B and Supplementary Data 14). We thus confirm previous results[48] and can exclude an involvement of the *bre* genes in MYb115-mediated protection against Bt247.

## Discussion

Understanding microbiota-host interactions at the level of the molecular mechanism requires the identification of individual microbiota-derived molecules and their associated biological activities that mediate the interaction. In this study we demonstrate that *P. fluorescens* MYb115-mediated host protection[18], depends on bacterial-derived SLs. We show that MYb115 produces protective SLs by a BGC encoding an iterative PKS and an AOS with SPT activity. This finding is important since eukaryotes and all currently known SL-producing bacteria depend on the serine palmitoyl transferase (SPT) enzyme, which catalyses the initial step in the de novo synthesis of ceramides, for SL production as primary metabolites. Indeed, the SPT gene is conserved between eukaryotes and prokaryotes and its presence in bacterial genomes has been used as an indication of SL production. While SL production is ubiquitous in eukaryotes, it is thought to be restricted to few bacterial phyla. Known SL-producing bacteria include the Bacteroidetes and Chlorobi phylum, and a subset of Alpha- and Delta-Proteobacteria[49]. More recently, two additional key enzymes required for bacterial ceramide synthesis have been identified, bacterial ceramide synthase and ceramide reductase[31]. Phylogenetic analysis of the three bacterial ceramide synthetic genes has identified a wider range of Gram-negative bacteria, as well as several Gram-positive Actinobacteria with the potential to produce SLs[31]. However, our finding that *P. fluorescens* MYb115 produces SLs by the BGC-encoded PKS/AOS *Pf*SgaAB, was previously unknown and therefore indicates that there are non-canonical ways of producing SLs as secondary metabolites in bacteria. Moreover, our analysis of the distribution of *Pf*SgaAB in bacteria revealed that homologous putative PKS clusters are present in bacteria that are so far unknown SL producers. This finding strongly suggests that PKS cluster-dependent biosynthesis of SLs is prevalent across bacteria.

By comparing the *C. elegans* transcriptome response to MYb115 with the response to the MYb115 PKS mutant in a metabolic network analysis, we observed an effect of MYb115-derived SLs on host fatty acid and SL metabolism. Our *C. elegans* lipidomic profiling corroborated the transcriptomic data, providing evidence that MYb115-derived SLs alter *C. elegans* SL metabolism, resulting in the reduction of certain complex SL species. A similar effect of gut microbiota-derived SLs on host lipid metabolism was previously observed in mice: *Bacteroides thetaiotaomicron*-derived SLs reduce de novo SL production and increase ceramide levels in the liver[37]. Also, *B. thetaiotaomicron*-derived SLs alter host fatty acid and SL metabolism and ameliorate hepatic lipid accumulation in a mouse model of hepatic steatosis[50]. In humans, bacterial SL production correlates with decreased host-produced SL abundance in the intestine and is critical for maintaining intestinal homeostasis[51]. Thus, interference with host SL metabolism may be a general effect of bacterial-derived SLs.

What role do MYb115-derived SLs play in host protection against Bt? The current study reveals that MYb115-derived SLs protect the *C. elegans* intestinal barrier, affect the activation of pathogen defence genes, and affect host fatty acid and SL metabolism. Consistent with an important role of host SLs in *C. elegans* defence, earlier studies have provided evidence for the involvement of SL metabolism in the host response to infection with *Pseudomonas aeruginosa* and *Enterococcus faecalis*[52,53]. We previously described an association between modulations in fatty acid and SL metabolism and increased tolerance

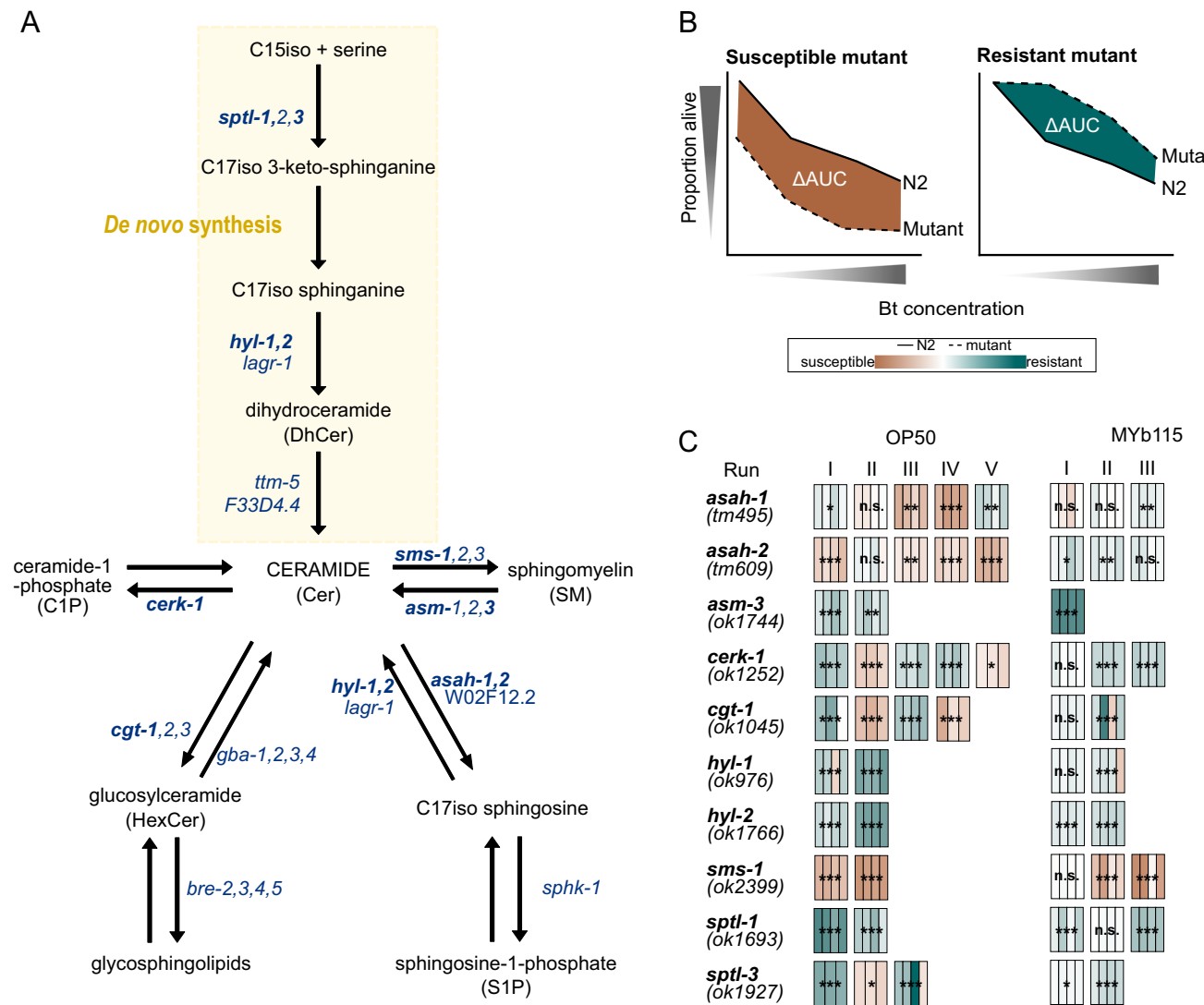

**Fig. 7 | Modulations in *C. elegans* SL metabolism affect survival after Bt247 infection. A** Overview of SL metabolism in *C. elegans*. *C. elegans* produces sphingoid bases which are derived from a C17 iso-branched fatty acid and are thus structurally distinct from those of other animals with mainly straight-chain C18 bases[40]. *C. elegans* SLs consist of a sphingoid base backbone derived from C15iso-CoA and serine, which is N-acylated with fatty acids of different lengths as well as different functional groups at the terminal hydroxyl group. Dihydroceramides (DhCers) are formed from C17iso sphinganine and fatty acids or 2-hydroxy fatty acids. Desaturation at the 4th carbon yields ceramides (Cers), which are the precursors of complex SLs such as sphingomyelin (SM) and glucosylceramide (Hex-Cer). Mutants of SL metabolism genes in bold were tested in survival assays shown in (**C**). **B** Schematic survival comparing N2 wildtype (solid line) *versus* mutant strains (dashed lines), the difference of the area under the survival curve (AUC) is

shaded in brown when the mutants are more susceptible to the infection than the control and in green when the mutants are more resistant to the infection. **C** Heatmap represents the ΔAUC of the survival of the *C. elegans* SL metabolism mutants *versus* average of the wildtype N2 strain. Each box represents an independent experiment, consisting of three to four technical replicates (individual bars). The intensity of the bar colour reflects the overall summary across all experiments, while the statistical analysis was performed separately for each experiment. Statistical analyses were carried out using the GLM framework with a binomial distribution. All tests were two-sided, and *p*-values were adjusted for multiple comparisons using the Bonferroni correction. Significance is indicated as *$p < 0.05$, **$p < 0.01$, ***$p < 0.001$. Each individual survival curve can be found in Fig. S17A, B. Source data and exact *p*-values are provided in Supplementary Data 12.

to Bt infection[48]. In line with this, we here demonstrate that modulations in SL metabolism strongly affect survival of infected animals. We do, however, not yet understand how exactly susceptibility to Bt infection is affected by SL modifications. Our functional genetic analysis of *C. elegans* SL metabolism enzymes shows that inhibition of de novo synthesis of ceramide and inhibition of the conversion of ceramide to glucosylceramides or ceramide-1-phosphate increases survival of *C. elegans* infected with Bt247, while inhibition of the conversion of ceramide to sphingomyelin or sphingosine decreases survival of Bt247-infected animals. Supplementation with C18 and C20 ceramide increased survival after Bt infection. These results and the phenotypic differences between the ceramide metabolic gene

mutants imply that the enhanced susceptibility to infection is influenced yet not exclusively caused by the inhibition of ceramide metabolism (and the associated increase in ceramide content) in these mutant backgrounds. The enzymes responsible for SL production and turnover comprise a complex metabolic network that gives rise to numerous bioactive molecules, which participate in highly complex and interconnected pathways influencing a multitude of physiological processes[54,55]. Also, SL metabolism shares common substrates with other metabolic routes and is, for example, highly connected to other lipid metabolic networks. Consequently, imbalances in SL metabolism in a mutant may have far-reaching consequences for host physiology.

MYb115 interacts with host SL metabolism at least at the conversion of ceramide to sphingomyelin and sphingosine, since the susceptibility phenotypes of the respective ceramide metabolic gene mutants are ameliorated or even abrogated in MYb115-treated animals, respectively. However, the effect of MYb115 on the phenotype of a SL metabolism mutant (increased survival after Bt infection) may not be directly linked to its effect on wildtype worms (decrease in sphingomyelin and other SLs). The conclusions we can draw from our data are that SL-producing MYb115 decreases certain host SL species, including sphingomyelin species, in comparison to non-SL-producing MYb115 and that MYb115 ameliorates the survival phenotype of *C. elegans* ceramide metabolic gene mutants following Bt infection. Indeed, we cannot exclude that this effect is indirect or due to other effects of MYb115 on the host.

SLs are not only required for the integrity of cellular membranes, but can also act as bioactive signalling molecules involved in regulation of a myriad of cell activities, including pathogen and stress defence pathways[54]. For example, many bacterial pathogens, including Bt, produce virulence factors that target and damage mitochondria[56,57]. A *C. elegans* surveillance pathway, which detects mitochondrial defects and activates xenobiotic-detoxification and pathogen defence genes, requires ceramide biosynthesis[46]. Since we, however, did not find any evidence of mitochondrial surveillance activation by Bt247 infection or MYb115 (Fig. S19A), it is unlikely that this defence pathway is involved in MYb115-mediated protection. Also, in *C. elegans*, glucosylceramide deficiency was linked to an increase in autophagy[58,59], which plays an important role in cellular defence after attack by certain Bt PFTs[60]. Notably, glucosylceramides serve as a source for the synthesis of complex glycosphingolipids. In *C. elegans*, the BRE proteins BRE-2, BRE-3, BRE-4, and BRE-5 are required for further glucosylation of glucosylceramide, leading to complex glycosphingolipids that are receptors of the *B. thuringiensis* Cry toxin Cry5B[47]. However, Bt247 only expresses the unique toxin App6Ba[61], which belongs to the PFT class of alpha helical pesticidal proteins[62]. These proteins are unrelated to Cry5B at the level of their primary sequences and structure[63]. Indeed, we could previously exclude an involvement of the *bre* genes in *C. elegans* defence against Bt247, given that *bre* mutants are susceptible to Bt247 infection (Fig. S19B[48]) and did not find evidence of an effect on MYb115-mediated protection (Fig. S19B). Still, MYb115-mediated interference with SL metabolism might affect membrane organisation and dynamics, as well as vesicular transport, which in turn might affect other membrane-associated Bt toxin receptors through modifying their localisation in the plasma membrane. *C. elegans* is thus an ideal experimental system to study the downstream impact of microbiota-derived SLs in the context of pathogen protection, an area that is still largely unexplored[64].

## Methods
### *C. elegans* strains and growth conditions
The wildtype *C. elegans* strain N2 (Bristol)[65] and all SL mutant strains were purchased as indicated in Table 1. Worms were grown and maintained on nematode growth medium seeded with the *E. coli* strain OP50 at 20 °C, according to the routine maintenance protocol[66]. Worm populations were synchronised and incubated at 20 °C. Since we did not cross out the mutant strains, we confirmed the mutations in all *C. elegans* sphingolipid mutant strains via PCR (Fig. S20). The following primer were used for genotyping: *asah-1*: forward AGTGGGTGTTCG-GATTGGAGG, reverse GGTTGGTGCGGGATGACAAG; *sptl-3*: forward AGCCGTGGCAAATGGAAAGTG, reverse ATGGAGTTTCTGCGCGATT GATG; *cgt-1*: forward ACTTCAGCTACCACTCCTTCATCAC, reverse AACTTTCCTTTCGATTCCTGGACC; *asah-2*: forward CGCCGAAGTGC TTGACGTAC, reverse CCAACATTGGCGCGAGTAAGC; *sms-1*: forward TGGTTGCGTTTCTGATGCTCG, reverse TGAGACCGAGCCCGAACATG; for *sptl-1*, *asm-3*, *hyl-1*, *hyl-2* and *cerk-1* already published primers were used[46] (Supplementary Data 15).

**Table 1 | Worm strains used in this study**

| Worm strain | Genotype | Origin |
|---|---|---|
| N2 | | CGC |
| RB1036 | *hyl-1(ok976)* | CGC |
| RB1498 | *hyl-2(ok1766)* | CGC |
| RB1487 | *asm-3(ok1744)* | CGC |
| RB1465 | *sptl-1(ok1693)* | CGC |
| RB1579 | *sptl-3(ok1927)* | CGC |
| RB1854 | *sms-1(ok2399)* | CGC |
| FX00495 | *asah-1(tm495)* | NBRP Tokyo Japan |
| FX00609 | *asah-2(tm609)* | NBRP Tokyo Japan |
| RB1203 | *cerk-1(ok1252)* | CGC |
| VC693 | *cgt-1(ok1045)* | CGC |
| GK70 | dkls37[*act-5p::GFP:pgp-1*] | 35 |
| FX03036 | *tir-1(tm3036)* | NBRP Tokyo Japan |
| AU3 | *nsy-1(ag3)* | CGC |
| KU4 | *sek-1(km4)* | Ewbank Lab |
| KU25 | *pmk-1(km25)* | CGC |
| KB3 | *kgb-1(ums3)* | CGC |
| SJ4100 | zcls13 [*hsp-6p::*GFP + *lin-15*(+)] | CGC |
| HY494 | *bre-2(ye31)* | Aroian lab |
| HY483 | *bre-3(ye26)* | Aroian lab |

### Bacterial strain and growth conditions
The standard laboratory food source *E. coli* OP50 was previously obtained from the CGC. The natural microbiota isolate *P. fluorescens* MYb115 (NCBI Reference Sequence: NZ_CP078138.1) isolated from the natural *C. elegans* strain MY379 was used[16].

The promoter-exchange strain MYb115 P$_{BAD}$*sga* for targeted in-/activation of the *sgaAB* BGC was generated via insertion of the inducible P$_{BAD}$ promoter upstream of the BGC following an established protocol[20]. The resulting plasmid (pCEP_kan_*sgaA*) was transformed into the conjugation host *E. coli* ST18 via electroporation and introduced into MYb115 via conjugation[67]. The promoter was induced by adding 0.02% (w/v) arabinose (ara) to the culture medium and repressed by adding 0.05% glucose (glc) to the growth medium. Deletions of the single genes *sgaA* and *sgaB* were carried out following a previously established protocol based on conjugation and homologous recombination[67,68]. Briefly, fragments upstream and downstream of the target gene were amplified by PCR and assembled into a plasmid using the pEB17 vector[69]. The resulting plasmids (pEB17_kan_Δ*sgaA* and pEB17_kan_Δ*sgaB*) were subsequently transformed into the conjugation host *E. coli* via electroporation and the plasmid was introduced into MYb115 via conjugation[67], sequences are shown in Supplementary Data 16. For the complementation of MYb115 Δ*sgaA* and MYb115 Δ*sgaB* we conducted a series of experiments, inserting the vanillic acid-inducible P$_{vanCC}$ promoter upstream of *sgaA* or the complete *sgaAB* BGC on the plasmid pSEVA631, which was then introduced into the MYb115 mutants. In detail: genomic DNA (gDNA) of MYb115 was isolated via Monarch® Genomic DNA Purification Kit (NEB) and used as the template for PCR amplification. The corresponding gene fragments of *sgaA* and *sgaAB*, together with the pSEVA631 (https://seva-plasmids.com/find-your-plasmid/) plasmid backbone with overhangs suitable for Gibson cloning, were amplified using Q5® High-Fidelity DNA Polymerase (NEB) and then purified by gel extraction using the Monarch® DNA Gel Extraction Kit (NEB). The following primers are used for PCR amplification: *sgaA* forward CTAGAGAAAGAGGGGAAA-TACTAGTTGACAAAGCGTAGACAGGTAG, *sgaA_reverse* CAGGGTTTT CCCAGTCACGACTCACTCAATCAAACGGTTAGGTG; *sgaAB* forward TTGACAAAGCGTAGACAGGTAG, *sgaAB_reverse* CAGGGTTTTCCCAGT CACGACTCACCCAATCTTCGCCAATTC; pSEVA631_backbone_forward

GTCGTGACTGGGAAAACCCT, pSEVA631_backbone_reverse CTAG-TATTTCCCCTCTTTCTCTAGT. Gibson cloning employing NEBuilder® HiFi DNA Assembly Cloning Kit (NEB) assembled the constructed plasmids, which were subsequently transformed by electroporation into electro competent *E. coli* DH10B. Finally, plasmids were isolated by the PureYield™ Plasmid Miniprep System (Promega).

Plasmids were transformed via electroporation into electro competent MYb115 mutants. At least three different colonies were selected for further small-scale production analysis. Cells were cultivated overnight in LB media with 75 µg/mL gentamicin. Afterwards, 100 µL overnight grown culture were inoculated in 5 ml XPP medium[69] containing 75 µg/mL gentamicin and 100 µM vanillic acid. The cells were cultivated for 3 days at 28 °C and 200 rpm.

One hundred microlitres of the cultures were taken and extracted with methanol at a 1:1 ratio by shaking 10 min at room temperature. Followed by further diluting the mixtures 1:10 with methanol and centrifuged at 13,000 rpm for 30 min. Cleared supernatants were used for further HPLC/MS analysis. HPLC/MS analysis was conducted on an UltiMate 3000 system (Thermo Fisher) coupled to an AmaZonX mass spectrometer (Bruker) with an ACQUITY UPLC BEH C18 column (130 Å, 2.1 mm × 100 mm, 1.7-µm particle size, Waters) at a flow rate of 0.4 mL/mL (5–95% acetonitrile/water with 0.1% formic acid, vol/vol,16 min, UV detection wavelength 190–800 nm) and an electrospray ionization (ESI) source set to positive ionization mode.

Only the complementation that included the complete *sgaAB* BGC restored SL production (detection of compounds **1** ($m/z$ 414.4 $[M + H]^+$), **2** ($m/z$ 386.4 $[M + H]^+$), and **3** ($m/z$ 442.4 $[M + H]^+$)) by HPLC/MS analysis and only in the MYb115 Δ*sgaA* mutant (Fig. S21). The resulting MYb115 Δ*sgaA*/P$_{vanCC}$*sgaAB* strain was tested in *C. elegans* survival assays, for which bacteria were first grown overnight at 28 °C with shaking (180 rpm) in 5 mL of Tryptic Soy Broth (TSB) supplemented with gentamicin (75 µg/mL). This was followed by a three-day cultivation in XPP medium[69] containing gentamicin and 100 µM vanillic acid to induce the P$_{vanCC}$ promoter. All other bacteria were grown on Tryptic Soy Agar (TSA) plates at 25 °C and liquid bacterial cultures were grown in TSB in a shaking-incubator overnight at 28 °C.

Of note: We could confirm targeted activation of SL production in MYb115 P$_{BAD}$*sga* by arabinose supplementation (Fig. 1C and https://metaspace2020.org/dataset/2025-02-27_13h37m58s), suggesting that the P$_{BAD}$ promoter is not leaky in this system. In contrast, SL production was observed in cultures of two MYb115 Δ*sgaA*/P$_{vanCC}$*sgaAB* strains even without induction by vanillic acid supplementation, indicating leakiness of the P$_{vanCC}$ promoter under certain conditions. However, the addition of vanillic acid usually led to a further significant increase in SL production (https://metaspace2020.org/dataset/2025-02-27_13h37m58s). All primer sequences can be found in Supplementary Data 1. For survival assays with *B. thuringiensis*, we used the strain MYBt18247, MYBt18679 (Bt247 and Bt679, respectively, our lab strains) and Bt407[70] as non-pathogenic control[48,71]. Spore aliquots of both strains were obtained following a previously established protocol[72] with minor modifications[18].

### Transcriptome analysis using RNA-seq
Roughly 500 synchronised N2 worms were raised on PFM plates inoculated with MYb115 or MYb115 Δ*sgaA* (OD$_{600nm}$ of 10) from L1 to L4 stage. At L4 stage worms were transferred to control plates or infection plates (microbiota mixed with Bt247 spores 1:100). Transcriptomic response was assessed 24 h post-transfer, with three independent replicates. Worms were washed off the plates with M9-T (M9 buffer + 0.02% Triton X-100), followed by three gravity washing steps. The worm pellets were resuspended in 800 µL TRIzol (Thermo Fisher Scientific, Waltham, MA, United States). Worms were broken up prior to RNA extraction by treating the samples with four rounds of freeze-thaw cycles using liquid nitrogen and a thermo block at 46 °C. The RNA

was extracted using Direct-zol™ RNA MicrolPrep (Zymo Research, R2062) and stored at −80 °C.

The RNA was processed by Lexogen (Vienna, Austria) using the 3′ mRNAseq library prep kit and sequenced on an Illumina NextSeq2000 on a P3 flow cell in SR100 read mode. FASTQ files were checked for their quality with MultiQC[73], filtered and trimmed with cutadapt[74] and aligned to the *C. elegans* reference genome WBcel235 with the STAR aligner (Spliced Transcripts Alignment to a Reference[75]) followed by an assessment using RseQC[76]. Ultimately, HTseq-count v0.6.0[77] generated the raw gene counts. The count normalization with the median of ratios method for sequencing depth and RNA composition as well as the analysis for differential expression by a generalised linear model (GLM) was performed using DESeq2[78]. Raw data and processed data have been deposited in NCBI's Gene Expression Omnibus[79] and are accessible through GEO Series accession number GSE245296.

### Liquid chromatography-mass spectrometry (LC-MS) analysis of MYb115
For LC-MS analysis, 1 mL liquid culture was harvested via centrifugation (1 min, 20 °C, 17,000 × $g$). The cell pellet was resuspended in 1 mL MeOH and incubated at 30 °C for 30 min. The resulting extract was separated from the cell debris via centrifugation (30 min, 20 °C, 17,000 × $g$), diluted and submitted to LC-MS measurements. LC-MS measurements were performed on a Dionex Ultimate 3000 (Thermo Fisher Scientific) coupled to an Impact II qToF mass spectrometer (Bruker Daltonics). Five microlitres sample were injected and a multi-step gradient from 5 to 95% acetonitrile (ACN) with 0.1% formic acid in water with 0.1% formic acid over 16 min with a flow rate of 0.4 mL/min was run (0–2 min 5% ACN; 2–14 min 5–95% ACN; 14–15 min 95% ACN; 15–16 min 5% ACN) on a Acquity UPLC BEH C18 1.7 µm column (Waters). MS data acquisition took place between minutes 1.5 and 15 of the multistep LC gradient. The mass spectrometer was set to positive polarity mode with a capillary voltage of 2.5 kV and a nitrogen flow rate of 8 L/min. We compared the MS$^2$ data of compounds **1**–**3** to the MS$^2$ data obtained from commercially available sphinganines (sphinganine (d18:0) and sphinganine (d20:0), Avanti Polar Lipids).

### Expression and purification of *Pf*SgaB
pET28a-*PfsgaB* (synthesised and cloned by Genscript) was used to transform chemically-competent *E. coli* BL21 (DE3) cells via the heat-shock method. Colonies were developed overnight on VLB-kanamycin agar plates (50 µg mL$^{−1}$). A single colony was propagated in LB-kanamycin media (50 mL) and incubated overnight (37 °C) with agitation. The cells were subcultured (OD$_{600nm}$ = 0.1, 37 °C) in fresh VLB-kanamycin media (500 mL) until mid-log phase. The cultures were cooled to room temperature and protein expression was induced by the addition of IPTG (0.1 mM). Protein expression proceeded overnight at 20 °C with rigorous agitation. The biomass was harvested by centrifugation using a Fiberlite F14-6 × 250y fixed-angle rotor (7000 rpm, 5 min), combined into 2–5 g yellow-tinged pellets using a Fiberlite F15-8 × 50cy fixed angle rotor (5000 rpm, 10 min) and stored at −20 °C. When needed, cell pellets were defrosted and resuspended (10% w/v) in ice-cold Binding/Storage buffer containing HEPES (50 mM, pH 7.5), NaCl (250 mM), glycerol (10% v/v) and PLP (25 µM). Benzamidine hydrochloride (1 mM) was added to the resuspension and the cells were lysed by sonication (10 s pulse/second cooldown, 15 cycles) on ice. Cell debris was pelleted by high-speed centrifugation using a Fiberlite F15-8 × 50cy fixed angle rotor (13,000 rpm, 40 min, 4 °C). The cell-free extract was collected and clarified by filtration (Millex-HP 0.45 µm polyethersulfone, Merck). *Pf*SgaB was purified from the cell-free extract using a HiTrap TALON Crude 1 mL column. The bound *Pf*SgaB protein was washed with copious Binding buffer (20 mL) and eluted with imidazole (150 mM). Yellow fractions containing *Pf*SgaB were pooled and further purified using a HiLoad Superdex S200 16/600 pg (120 mL) SEC column, using Binding/

Storage buffer as mobile phase. Purified fractions were concentrated by centrifugal concentration (50 kDa MWCO, <4 mL). For long-term storage, aliquots of $Pf$SgaB were flash frozen in liquid nitrogen and stored at −80 °C.

## Detection of external aldimine formation by UV–vis

A reaction mixture (1000 μL) containing purified $Pf$SgaB (10 μM) and L-serine (0.3125–10 mM) was prepared in a reaction buffer containing HEPES (50 mM, pH 7.5), NaCl (250 mM) and glycerol (10% v/v). The mixture was incubated at room temperature for 5 min and analysed using a pre-blanked spectrophotometer (300–700 nm). External aldimine $\lambda_{max}$ = 413 nm.

## DTNB activity assay

A reaction mixture (200 μL) containing purified $Pf$SgaB (5 μM), L-serine (10 mM) and DTNB (250 μM) was initiated by the addition of acyl-CoAs **7**–**9** (100 μM) in a reaction buffer containing HEPES (50 mM, pH 7.5), NaCl (250 mM) and glycerol (10% v/v). Negative controls were prepared by the replacement of the reaction component(s) with buffer. Amino acid specificity was determined by the replacement of L-serine with L-alanine (10 mM) or glycine (10 mM). Absorbances were measured over the course of 20–60 min using a BioTek Synergy HXT (28 °C, 412 nm), configured for pathlength correction. A molar attenuation coefficient of 14150 $M^{-1}$ $cm^{-1}$ was used to convert absorbance into concentration using Beer's law.

## $Pf$SgaB-catalysed 3-KDS formation using LC/ESI-MS

A reaction mixture (200 μL) containing purified $Pf$SgaB (10 μM), L-serine (10 mM) and acyl-CoA (100 μM) was prepared in a reaction buffer containing HEPES (50 mM, pH 7.5), NaCl (250 mM) and glycerol (10% v/v). The reactions were incubated at 28 °C for 18 h with rigorous shaking. The reactions were quenched by the addition of ice-cold LC-MS grade MeOH (200 μL) containing formic acid (2% v/v). Precipitate was removed by microcentrifugation (13,300 rpm, 5 min). The supernatant was sampled for LC/ESI-MS analysis in positive ion mode using a Waters SYNAPT G2 HDMS, equipped with a Waters ACUITY Premier CSH C18 column (1.7 μm particle size, 2.1 mm ID, 100 mm length). Analytes were resolved using a water/ACN gradient (5–95% ACN) over 12 min. 0.1% formic acid was used as the mobile phase modifier.

## Bioinformatics

The *P. fluorescens* MYb115 (NCBI accession: NZ_CP078138) SL BGC was identified and annotated using antiSMASH[80]. Sequence homologues were retrieved using BLASTp and UniProt. Multiple sequence alignments were generated using ClustalOmega[81] and visualised using ESPript 3.0[82]. AlphaFold3[83] was used for predictive structural modelling. Structural models were visualised and analysed using ChimeraX (v1.8)[84].

## Labelling experiments

Bacterial cultures producing the sphinganine compounds were grown in ISOGRO®-13C and ISOGRO®-15N (Sigma Aldrich) medium and subsequently analysed by LC-MS to determine the number of carbon and nitrogen atoms, respectively. To confirm the incorporation of serine into the sphinganines, MYb115 $P_{BAD}sga$ cultures were grown in XPP medium[69] with addition of all proteinogenic amino acids (Carl Roth GmbH + Co. KG, Karlsruhe) except serine. To test the incorporation, either $^{13}C_3^{15}N$-labelled (Sigma Aldrich) serine or regular serine (Carl Roth GmbH + Co. KG, Karlsruhe) displaying the usual isotopic abundances were used. This should result in the production of two isotopologues of each sphinganine. With addition of $^{13}C_3^{15}N$-labelled serine, the isotopologue that is $m_{monoisotopic}$ + 3 should be labelled with two $^{13}C$ isotopes and one $^{15}N$ isotope, since one carbon atom is lost through the elimination of $CO_2$ during the condensation. In the cultures with regular serine, the isotopologue that is $m_{monoisotopic}$ + 3 should be labelled with three $^{13}C$ isotopes because of the higher natural abundance of $^{13}C$ compared (1.1%) to $^{15}N$ (0.4%). The two isotopologues, $^{13}C_3$ and $^{13}C_2^{15}N$, were distinguished by their respective masses.

## Metabolic modelling

For the metabolic model analysis, transcriptomic data was integrated into the iCEL1314 *C. elegans* metabolic model using the MERGE pipeline[39] in MATLAB (version: 9.11.0.1769968 (R2021b)) using the COBRA toolbox[85]) to create context-specific (CS) models of each sample using iMAT++[27]. This method not only integrates transcriptomic data into the model, but also simulates the optimal flux distribution (OFD) for the fitted transcriptomic data, as well as provides a flux variability analysis (FVA)[41] output that describes the minimum (lb) and maximum (ub) flux values that each reaction can take within each CS model under the same in silico dietary conditions. Gene categorization was performed in Python[86] (version 3.10.6) using 0.7816 (mu1), 4.856 (mu2) and 8.15 (mu3), as rare, low, and high expression category cutoffs, respectively. We would like to point out that the iMAT++ algorithm used to integrate the transcriptomic data into the iCEL1314 metabolic model is done on a sample basis, therefore any statistical comparisons of gene expression are not taken into account during this process. This means that comparing the differences in simulation results of reactions encoded by a certain gene and the logFC values of this gene might not directly match in direction. This is a desirable attribute of metabolic modelling since we can predict metabolic requirements of an organism that are in conflict with the gene expression differences—these might be caused by post-translational modifications or other effects. Differences between generated metabolic models were assessed by fitting a linear regression model (data ~ treatment) using FVA[41] centres (([ub-lb]/2)) and OFD values (equivalent to parsimonious FBA solution) from each model. We subsequently contrasted MYb115 and MYb115 $\Delta sgaA$, combining unique significant reaction names (alpha = 0.01) across the different simulation data layers (OFD and centres), and performed a Flux Enrichment Analysis (FEA)[61] using these names to obtain significantly affected metabolic model pathways. For SL metabolism pathway analysis, FVA was performed on all reactions, with biomass objective minimum set to 50%. Upper bound values were grouped by pathway, then normalized against the mean on the MYb115 flux values for each reaction. Lower bound values were not analysed due to the unidirectional nature of most reactions (lb = 0).

## *P. fluorescens* MYb115 lipidomics

For the bacterial lipidomics experiment, we adapted the extraction method from Brown et al. [51]. 5 mL liquid cultures were incubated for 24 h at 30 °C. The equivalent of 1 mL $OD_{600nm}$ of 5 was harvested by centrifugation (1 min, 20 °C, $17,000 \times g$). The cell pellet was resuspended in 0.4 mL $H_2O$. 1.5 mL $CHCl_3$/MeOH (1:2) were added and the extracts were mixed by vortexing. The cell mixture was incubated at 30 °C with gentle shaking, after 18 h 1 mL $CHCl_3/H_2O$ (1:1) was added. After phase separation, the organic phase was dried using a nitrogen evaporator and stored at −20 °C.

The relative quantification and annotation of lipids was performed by using HRES-LC-MS/MS. The chromatographic separation was performed using a Acquity Premier CSH C18 column (2.1 × 100 mm, 1.7 μm particle size, VanGuard) a constant flow rate of 0.3 mL/min with mobile phase A being 10 mM ammonium formate in 6:4 ACN:water and phase B being 9:1 IPA:ACN (Honeywell, Morristown, New Jersey, USA) at 40 °C. For the measurement, a Thermo Scientific ID-X Orbitrap mass spectrometer was used. Ionisation was performed using a high temperature electrospray ion source at a static spray voltage of 3500 V (positive) and a static spray voltage of 2800 V (negative), sheath gas at 50 (Arb), auxiliary gas at 10 (Arb), and ion transfer tube and vaporiser at 325 and 300 °C, respectively.

Data dependent MS² measurements were conducted applying an orbitrap mass resolution of 120,000 using quadrupole isolation in a mass range of 200–2000 and combining it with a high energy collision dissociation (HCD). HCD was performed on the ten most abundant ions per scan with a relative collision energy of 25%. Fragments were detected using the orbitrap mass analyser at a predefined mass resolution of 15,000. Dynamic exclusion with an exclusion duration of 5 s after 1 scan with a mass tolerance of 10 ppm was used to increase coverage. For lipid annotation, a semi-quantitative comparison of lipid abundance and annotated peaks were integrated using Compound Discoverer 3.3 (Thermo Scientific). The data were normalised to the maximum peak area sum of all samples, the p-value per group ratio calculated by a one-way ANOVA with Tukey as post-hoc test, and the p-value adjusted using Benjamini-Hochberg correction for the false-discovery rate[87]. The p-values were estimated by using the log-10 areas. The normalized peaks were extracted and plotted using R (4.1.2) within RStudio using the following packages: ggplot2 (3.4.0), readxl (1.4.1), grid (4.1.2), gridExtra (2.3) and RColorBrewer (1.1-3). Metabolomics data have been deposited to the EMBL-EBI MetaboLights database[88] with the identifier MTBLS8694.

### PKS distribution analysis

The monomodular PKS (KW062_RS19805) and the AOS aminotransferase (KW062_RS19800) in *P. fluorescens* MYb115 (NZ_CP078138) were searched against the non-redundant (nr) National Center for Biotechnology Information (NCBI) database using cblaster (1.3.18)[89]. PKS clusters encoded by various bacterial genera were aligned and visualised using clinker[90].

### *C. elegans* lipidomics

For lipidomic profiling, N2 worms exposed to MYb115 or MYb115 Δ*sgaA* were used. Approximately 10,000 worms were raised on either of the bacteria for 70 h until they were young adults. Excess bacteria were removed by three gravity washing steps using M9 buffer. The buffer was thoroughly removed, and the samples were snap-frozen in liquid nitrogen.

Extraction and analysis of lipids were performed as described previously[91]. Worm pellets were suspended in MeOH and homogenised in a Precellys Bead Beating system (Bertin Technologies, Montigny-le-Bretonneux, France), followed by addition of MTBE. After incubation water was added and through centrifugation the organic phase was collected. The aqueous phase was re-extracted using MTBE/MeOH/$H_2O$ (10/3/2.5 v/v/v). Organic phases were combined and evaporated to dryness using a SpeedVac Savant centrifugal evaporator (Thermo Scientific, Dreieich, Germany). Proteins were extracted from the residue debris pellets and quantified using a BCA kit (Sigma-Aldrich, Taufkirchen, Germany). Lipid profiling was performed using a Sciex ExionLC AD coupled to a Sciex ZenoTOF 7600 under control of Sciex OS 3.0 (Sciex, Darmstadt, Germany). Separation was achieved on Waters Cortecs C18 column (2.1 mm × 150 mm, 1.6 μm particle size) (Waters, Eschborn, Germany). 40% $H_2O$/60% ACN + 10 mM ammonium formate/0.1% formic acid and 10% ACN / 90% iPrOH + 10 mM ammonium formate/0.1% formic acid were used as eluents A and B. Separation was carried out at 40 °C at a flow rate of 0.25 mL/min using a linear gradient as followed: 32/68 at 0.0 min, 32/68 at 1.5 min, 3/97 at 21 min, 3/97 at 25 min, 32/68 at 25.1 min, 32/68 at 30 min. Analysis was performed in positive ionisation mode.

Dried samples were re-dissolved in $H_2O$/ACN/iPrOH (5/35/60, v/v/v) according to their protein content to normalise for differences in biomass. Ten microlitres of each sample were pooled into a QC sample. The remaining sample was transferred to an autosampler vial. The autosampler temperature was set to 5 °C and 5 μL were injected for analysis. MS¹ ions in the m/z range 70–1500 were accumulated for 0.1 s and information dependent acquisition of MS² was used with a maximum number of 6 candidate ions and a collision energy of 35 eV with a

spread of 15 eV. Accumulation time for MS² was set to 0.025 s yielding a total cycle time of 0.299 s. ZenoTrapping was enabled with a value of 80,000. QC samples were used for conditioning of the column and were also injected every 5 samples. Automatic calibration of the MS in MS¹ and MS² mode was performed every 5 injections using the ESI positive Calibration Solution for the Sciex X500 system or the ESI negative Calibration Solution for the Sciex X500 system (Sciex, Darmstadt, Germany).

Data analysis was performed in a targeted fashion for SLs (Supplementary Data 11). SLs were identified by manual interpretation of fragmentation spectra following established fragmentation for different SL classes: m/z 268.263491, 250.252926 and 238.252926 for C17iso sphingosine and m/z 270.279141, 252.268577 and 288.289706 for C17iso sphinganine based derived SLs. Data analysis was performed in Sciex OS 3.0.0.3339 (Sciex, Darmstadt, Germany). Peaks for all lipids indicated below were integrated with a XIC width of 0.02 Da and a Gaussian smooth width of 3 points using the MQ4 peak picking algorithm. All further processing was performed in R 4.2.1 within RStudio using the following packages: tidyverse (v1.3.2), readxl (1.4.1), ggsignif (0.6.4), ggplot2 (3.3.6), scales (1.2.1). Significance was tested using a two-sided Welch-Test within ggsignif. Metabolomics data have been deposited to the EMBL-EBI MetaboLights[88] database with the identifier MTBLS8440.

### Bacterial SL peak intensity

Bacterial biomass was screened for the presence of SLs via MALDI mass spectrometry spot assays. Briefly, 1 μl of each cell pellet was transferred on a microscopy slide and air dried. The microscopy slide with spots of cells was covered with a matrix (sDHB) using a pneumatic sprayer (HTX Science). The spots were analysed using the AP-SMALDI5 AF source (Transmit, Giessen) connected to a QExactive HF (Thermofisher) as described previously[22].

A step size of 100 μm in X and Y direction was used to image all spots in positive ionisation mode. The MS settings where as followed: positive ionisation mode, mass range m/z 300–1200 Da, S-Lens 100, capillary voltage 4 kV, mass resolution 240000 at m/z 200 Da. The raw data was uploaded to figshare https://doi.org/10.6084/m9.figshare.29093192.v1 and transformed into imzml and deposited to metaspace2020.org for browsing of images (datasets: MPIMM_514_QE_P https://metaspace2020.org/dataset/2025-02-27_13h37m58s). For relative quantification of each compound **1–6**, a region of interest (ROI) was drawn around each bacterial spot and the peak intensity for M + H + (see Supplementary Data 1) per pixel was averaged within this ROI.

### Bt survival assay

*B. thuringiensis* survival assays were performed as described previously with minor adjustments[18,92,93]. N2 wildtype worms and the SL mutants were synchronised and grown on PFM plates seeded with 1 mL MYb115 or OP50 (OD$_{600nm}$ of 10) until they reached the L4 stage. Infection plates were inoculated with each of the bacteria adjusted to OD$_{600nm}$ of 10 mixed with Bt247 spores or Bt407. For the infection L4 worms were washed off the plates with M9 buffer and 30 worms were pipetted onto infection plates and incubated at 20 °C. To assess survival, all worms were counted as either alive or dead 24 h after infection. Worms were considered dead if they did not respond to light touch with a platinum wire picker. We plotted all survivals as survival curves (Fig. S17) but provided a summary of the data in a heatmap (Fig. 7C). The area under the survival curve (AUC) was calculated for the *C. elegans* mutant strains and the mean AUC of *C. elegans* wildtype N2. The AUC for the mutant strain was then subtracted from the mean AUC of wildtype worms (ΔAUC). Based on the ΔAUC values, the shading for the heatmap was determined (Fig. 7B). To test the effect of SLs on the survival of *C. elegans* we supplemented the worms with a range of different commercially available SLs. C18, C20 and C22 ceramide, as well as C16 and C18 sphingomyelin, were prepared in ethanol at

concentrations of 0.5 mg/ml or 10 mg/ml, respectively. Prior to inoculating the Bt assay plates with Bt-OP50 mixture, 60 μg of the SLs were inoculated onto PFM plates and thoroughly dried. A stock solution of 25 mM of D-Sphingosine in EtOH was prepared, for the assay either 50 or 100 μM were used for each infection plate. Sphingosine-1-phosphate was diluted in MeOH (2 mM) and 20 μM was used for each infection plate. In all survival assay equal amounts of EtOH or MeOH was used as control treatment. All SLs were obtained from Biomol GmbH - Life Science Shop, Germany.

Bt survival assays were done each with three to four replicates per treatment group and around 30 worms per replicate for each independent experiment. Statistical analyses were performed with RStudio (Version 4.1.2)[94]. GLM analysis with Tukey multiple comparison tests[95] and Bonferroni[96] correction were used for all survival assays individually. Graphs were plotted using ggplot2[97] and were edited in Inkscape (Version 1.4).

### Bacterial colonisation assay

To test for differences in colonisation of *C. elegans* L4 and young adults by MYb115 and MYb115 Δ*sgaA*, colonisation was quantified by counting colony forming units (CFUs). Worms were exposed to MYb115 and MYb115 Δ*sgaA* from L1 to L4 larval stage or additionally 24 h until worms reached young adulthood. To score the CFUs, worms were washed off their plates with M9-T (M9 buffer + 0.025% Triton-X100) followed by five gravity washing steps with M9-T. Prior to soft bleaching, worms in M9-T were paralysed with equal amounts of M9-T and 10 mM tetramisole to prevent bleach solution entering the intestine. Worms were bleached for 2 min with a 2% bleach solution (12% NaClO diluted in M9 buffer). Bleaching was stopped by removing the supernatant and washing the samples with PBS-T (PBS: phosphate-buffered saline + 0.025% Triton-X100). A defined number of worms was transferred into a new tube with PBS-T. A subsample of this was used as a supernatant control, while the remaining sample was homogenised with sterile zirconia beads (1 mm) using the BeadRuptor 96 (omni International, Kennesaw Georgia, USA) for 3 min at 30 Hz. Homogenised worms were diluted (1:10/1:100) and plated onto TSA plates, as well as the undiluted supernatant as control. After 48 h at 25 °C, colonies were counted and the CFUs per worm were calculated. To determine significant differences, we performed a *t*-test.

### Pumping behaviour

To score the pumping rate, i.e., the back and forth movement of the grinder, worms were exposed to either MYb115 or MYb115 Δ*sgaA*. Pumping was scored at L4 larval stage, young adults and young adults infected with Bt247 (1:100). Only worms that were on the bacterial lawn were counted for a period of 20 s. 15–20 worms per condition were counted. To determine significant differences, we performed pairwise Wilcoxon test.

### *C. elegans* intestinal integrity

To visualise intestinal morphology and integrity of *C. elegans* upon infection with Bt247 the worm strain GK70 (dkls37[P*act-5*::GFP:*pgp-1*]) was synchronized. L1 larvae were exposed for 72 h to *E. coli* OP50, *P. fluorescens* MYb115 Δ*sgaA* and MYb115 P$_{BAD}$*sga* supplemented with arabinose, followed by a 4 h infection with Bt247. Worms were picked into 10 mM tetramisole on agar-padded object slides. The number of membrane vesicles was counted and sorted into either of the three categories: "0 vesicles", "≤ 10 vesicles" or "> 10 vesicles". For each treatment populations of 14–25 worms were scored, the experiment was repeated three times.

### Reporting summary

Further information on research design is available in the Nature Portfolio Reporting Summary linked to this article.

## Data availability

*Caenorhabditis elegans* RNAseq data reported in this article have been deposited at NCBI's Gene Expression Omnibus and are accessible through GEO Series accession number GSE245296. *P. fluorescens* MYb115 metabolomics data have been deposited to the EMBL-EBI MetaboLights database with the identifier MTBLS8694. *C. elegans* metabolomics data have been deposited to the EMBL-EBI Metabolights database with the identifier MTBLS8440. The raw data of the MALDI mass spectrometry spot assay was transformed into imzml and uploaded to figshare https://doi.org/10.6084/m9.figshare.29093192.v1 and deposited to metaspace2020.org for browsing of images (datasets: MPIMM_514_QE_P https://metaspace2020.org/dataset/2025-02-27_13h37m58s). Source data supporting the findings of this study are provided with this paper and its Supplementary Data.

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

## Acknowledgements

We thank Lena Bluhm, Laura Brügmann, Hanne Griem-Krey, Sabrina Butze and Miriam Sadowski for their technical support. This work was funded by the German Science Foundation DFG (Collaborative Research Center CRC1182 Origin and Function of Metaorganisms, project A1.2 to K.D., project A1.1 to H.S., project A1.5 to C.K. and project Z to M.L.). DJC thanks the Biotechnology and Biological Sciences Research Council (BBSRC) for funding (grant number: BB/Y002210/1). We thank the Cae-norhabditis Genetics Center (University of Minnesota, Minneapolis, Minnesota, USA), funded by the NIH Office of Research Infrastructure Programs (P40OD010440) for *C. elegans* strains. Work in the Bode lab was partially supported by an ERC advanced grant (835108) and the Max-Planck Society. Work in the Metabolomics and Proteomics Core and Research Unit Analytical BioGeoChemistry, Helmholtz Zentrum München was partially supported by the German Science Foundation DFG (Project number 431572533 (MetClassNet) to M.W.).

## Author contributions

L.P., H.S., D.J.C., H.B.B. and K.D. conceptualized the project. L.P., M.D., M.A.H., J.L., B.P., J.J., A.C., F.L. and L.S. performed experiments. L.P., M.D., M.A.H., J.L., B.P., J.J., G.A., K.A.M., Y.M.S., M.L. and M.W. analysed data. N.P., H.S., C.K., M.W., M.L., D.J.C., H.B.B. and K.D. supervised the work. L.P. and K.D. interpreted the data and wrote the initial draft of the manuscript with support from M.D., M.A.H., K.A.M. and M.W. All authors discussed and revised the manuscript.

## Funding

## Competing interests

The authors declare no competing interests.

## Additional information

[1]Department of Evolutionary Ecology and Genetics, Zoological Institute, Kiel University, Kiel, Germany. [2]Department of Natural Products in Organismic Interactions, Max-Planck-Institute for Terrestrial Microbiology, Marburg, Germany. [3]Molecular Biotechnology, Department of Biosciences, Goethe-University Frankfurt, Frankfurt, Germany. [4]School of Chemistry, The University of Edinburgh, Edinburgh, UK. [5]Core Facility for Metabolomics and Small Molecule Mass Spectrometry, Max Planck Institute for Terrestrial Microbiology, Marburg, Germany. [6]Research Unit Analytical BioGeoChemistry, Helmholtz Zentrum München, Neuherberg, Germany. [7]Research Group Medical Systems Biology, Institute for Experimental Medicine, Kiel University, Kiel, Germany. [8]CAS Key Laboratory of Quantitative Engineering Biology, Shenzhen Institute of Synthetic Biology, Shenzhen Institute of Advanced Technology, Chinese Academy of Sciences, Shenzhen, China. [9]Max Planck Institute for Evolutionary Biology, Plön, Germany. [10]Metabolomics and Proteomics Core, Helmholtz Zentrum München, Neuherberg, Germany. [11]Chair of Analytical Food Chemistry, TUM School of Life Sciences, Technical University of Munich, Freising-Weihenstephan, Germany. [12]Department of Metabolomics, Institute for Human Nutrition and Food Science, Kiel University, Kiel, Germany. [13]Max Planck Institute for Marine Microbiology, Bremen, Germany. [14]Center for Synthetic Microbiology (SYNMIKRO), Phillips University Marburg, Marburg, Germany. [15]Department of Chemistry, Phillips University Marburg, Marburg, Germany. [16]Senckenberg Gesellschaft für Naturforschung, Frankfurt, Germany. [17]These authors contributed equally: Dominic J. Campopiano, Helge B. Bode, Katja Dierking. ✉e-mail: Dominic.Campopiano@ed.ac.uk; helge.bode@mpi-marburg.mpg.de; kdierking@zoologie.uni-kiel.de

