## [Transparent Peer Review file · Nature Communications]

Polyketide synthase-derived sphingolipids mediate microbiota protection against a bacterial pathogen in *C. elegans*

Corresponding Author: Dr Katja Dierking

Version 0:

Reviewer comments:

Reviewer #1

(Remarks to the Author)

This is an important study characterizing a polyketide synthase from *Pseudomonas fluorescens* MYb115, a microbe associated with *C. elegans* in nature that has been shown to offer *C. elegans* protection from pathogenic Bt infections. The researchers identified the polyketide synthase (PKS) activity, deleted it, and found that the microbe no longer offers protection against Bt infection in *C. elegans*. They used LC-MS to identify the product of the polyketide synthase, and found that the enzyme produces long-chain sphinganine molecules and then used isotopic labeling to show that serine is incorporated into the sphinganine molecules allowing them to propose a pathway for the activities of the PKS. This is very solid work and is important and interesting because the researchers also identified genes encoding PKS activity in many other microbes, especially those who interact with host organisms. This is a significant finding because the PKS were previously believed to be found in fungi and rarely in bacteria.

The investigators then sought to determine how the PKS products protect *C. elegans* from Bt pathogens. Sphingolipids are important components of the intestinal barrier, so the researchers used transcriptomic and lipidomic analysis to determine changes in the *C. elegans* sphingolipids in worms eating the wild type bacteria vs the bacteria in which the PKS was deleted. Interestingly, they found that the *C. elegans* that were colonized by the wild type MYb115 bacteria had lower levels of many sphingolipid species compared to worms who were colonized by the mutant bacteria lacking PKS. However, this may be minor because only a few species were significantly changed. Researchers then knocked down sphingolipid metabolism genes in *C. elegans* and examined susceptibility to Bt. They found that knocking down enzyme in the de novo pathway leading to ceramide protected worms from infection, while knocking down sphingomyelin synthase, which converts ceramide to sphingomyelin, resulted in more susceptible worms. The MYb115 somewhat rescued this susceptibility of the sms-1 knockout, although this is puzzling, because the researchers showed that colonization by MYb115 leads to reduced sphingomyelins in worms, so it is unclear how this rescue occurred.

This paper is well written and the data are clearly and thoroughly presented. While the *C. elegans* experiments did not clarify how the sphinganine products from MYb115 lead to protection from Bt infection in *C. elegans*, the observations are important starting points for future experiments. The researchers used rather vague terms, for example on line 309 "These data indicate that MYb115 interacts with host sphingolipid metabolism....." While the nature of the interaction isn't clear, at this point I think this is a valid statement, because more definitive mechanism can't be determined from the data. One technical comment. When researchers receive deletion lines from the CGC (ok alleles) or NBRP (tm alleles) they typically perform outcrosses against wild type worms since the original deletion strains can contain many other mutations. The researchers should report how many times the strains were outcrossed prior to their experiments, or else say that they were not outcrossed.

Reviewer #2

(Remarks to the Author)

The manuscript Peters et al. identified an iterative type I polyketide synthase (PKS) in MYb115 and showed that PKS regulates bacterial sphingolipid biogenesis and this interferes with sphingolipid metabolism of the host, thus exerting a

protective role for the host against pathogen infection. The paper contains a number of interesting observations, especially identification of an SPT-independent mechanism for sphingolipid biogenesis. However, the data did not provide a significant novelty and new insights into mechanistic explanation for microbiota-host interaction in regulation of host immune response, given that the role of sphingolipid in immune response and the interplay between bacterial sphingolipid and host sphingolipid biogenesis/metabolism are known. In addition, some of the claims are also not sufficiently supported by the results. For these reasons, I am not recommending publication in Nature Communications.

Major issues:

1. *sgaAB* encoded PKS catalyses sphingolipid biosynthesis and improves resistance of animals against Bt247.

- a) Is the protective effect of MYb115 linked with altered activity of known pathways for worm innate immunity, e.g. p38, MAPK, JUN-1, ELT-2, necrosis pathway or *bre* genes?
- b) How does MYb115 affect toxicity of the other Gram-positive pathogen, e.g. Bt679?
- c) Can the MYb115 Δ *sgaAB* phenotype be rescued by restoring *sgaAB* expression in this mutant bacterium?
- d) Can expression of *sgaAB* in another bacterium which normally doesn't have a protective role against Bt247 result a protection?
- e) How is the intestine morphology and integrity upon Bt247 infection affected by MYb115 and *sgaAB*?
- f) Line 146: MYb115 lipidomic analysis was performed to test whether the bacteria sphingamines exist as free compounds or are part of lipids and sphingamines 1-3 and PG-sphingolipids 4-6 were identified. Does this result suggest that the bacteria sphingamines exist as both free compounds and part of lipids? A clear conclusion to the question should be provided to make understanding easier, especially for those who are not familiar with sphingolipid metabolism. In addition, I don't understand why elucidating the existing form of sphingamines is important to know how MYb115 interact with the host. Additional background explanations are required to understand the underlying logic. The author concluded that they are not able to differentiate between the effects of the individual sphingolipid species. I don't think that lipidomic analysis alone would help to answer this question. But supplementation of these sphingamines or PG-sphingolipids might do.

2. Transcriptome study:

- a) It is surprising that Bt247 infection only cause few numbers of DEG in MYb115 and worms on the MYb115 Δ *sgaAB* mutants, given their significantly different survival rate. As the RNA-seq results was not shown in any main figures or supplemental data, except an enrichment analysis in Fig. 4A and S4A, it is difficult to make much comments. Does Bt247 infection affect transcriptome significantly? Does it result in activation of innate immunity genes? If Bt247 does not cause a significant alteration in gene expression, it would be hard to find involved factors via comparing transcriptome of animals on MYb115 and on the MYb115 Δ *sgaAB* mutants. In a such scenario, MYb115 might protect animals via other mechanisms that are independent of influencing transcript level. How does MYb115 affect transcriptome without Bt247 infection? If the Bt247 infectious bacteria remodel the worms' transcriptome much more than MYb115 does, the the potential changed genes that are only influenced by MYb115 or MYb115 Δ *sgaAB* mutants might submerge among Bt247-caused alteration in gene expression.
- b) The transcriptome data was then integrated into the iCEL1314 genome-scale metabolic model and 24 and 23 significant differences in the presence or absence of Bt247 were received. What does "24 and 23 significant differences" mean? Does it refer to Differences between generated metabolic models? The results must be described in more details so that non-experts in iCEL1314 could understand the data. What does the "Gene Ratio" in the Figure 4A mean? Fig. 4A and S4A only show enrichment of metabolic processes even without notifying whether these processes are activated or inactivated by *sgaAB* in MYb115 (whether genes involved are up- or down-regulated). A list of genes used for the enrichment analysis would also be important to understand the result.
- c) Line 220: "Here, we also saw an enrichment in valine, leucine, and isoleucine degradation, which are branched-chain amino acids (BCAA). This pathway is directly connected with propanoate metabolism that provides components for the synthesis of the C15iso fatty acid, which is the precursor for sphingolipids in *C. elegans*": are these genes up- or down-regulated in MYb115 vs. worms on the MYb115 Δ *sgaAB* mutants? Again, provide more RNA-seq information will avoid such a confusion.
- d) The RNA-seq data should be validated via qRT-PCR.

3. Host sphingolipid metabolism interferes with Bt247 susceptibility

- a) From the host lipidomics analysis shown in Fig. 4b, *sgaAB* in MYb115 seems to reduce level of Cer, HexCer, DhCer and SM in the host. To test whether accumulation of these molecules reduces survival of animals upon Bt247 infection, the authors used different mutants in the sphingolipid metabolism pathway to test their susceptibility to Bt247. However, *cgt-1* and *cerk-1* mutant animals fed with OP50 show increased survival, although these animals are supposed to have Cer accumulation. In contrast, *asm-3*, *hyl-1*, *hyl-2*, *sptl-1* and *sptl-3* mutants which probably have reduced Cer level are also resistant. Do these results argue against an important role of Cer in Bt247 susceptibility? In summary, from these Bt247 survival assay with different mutants in sphingolipid metabolism pathway, the author could not provide a clear answer which classes of host sphingolipid are responsible for the altered susceptibility to the pathogen.
- b) What is account for Bt247 toxicity? Why should altered sphingolipid metabolism influence toxicity of Bt247? A study from Ruvkun's lab (Liu et al., 2014, Nature) has shown that some nature habitat bacteria cause mitochondria dysfunction and animals respond with mitochondrial surveillance machinery which is ceramide dependent. At the same time some other bacteria could inhibit mitochondrial surveillance to render a more effective virulence. Does Bt247 modulate the mitochondria surveillance? If yes, could *sgaAB* in MYb115 counteract the Bt247 effect on mitochondria?

4. Sphingolipid in MYb115 affect host sphingolipid biogenesis and metabolism.

There is no attempt to address how this could happen. Is uptake of the bacterial sphingolipid into worm intestine necessary? Can feeding worms with the identified sphingamines 1-3 or sphingolipid 4-6 impact Bt247 resistance? If yes, does this supplementation change sphingolipid composition of worms?

Minor issues

1. Zeile 377: Notably, in *C. elegans*, glucosylceramide deficiency was linked to an increase in autophagy which plays an important role in cellular defence after attack by certain Bt pore-forming toxins (PFTs). Does Myb affect autophagy? If yes, could manipulating autophagy affect survival of animals to Bt247?
2. Fig. S6: Should A1 be compared with B1, A2 with B2 and so on? If yes, why is the mutant A10 in red line and B10 in yellow? The color used for asah-1 and asah-2 are too similar to be differentiated. Maybe just add the name of the mutant directly to the figures. Why are there several figures for some mutants, e.g. asah-1 while for some others only one e.g. cgt-1 on OP50? It does not seem to be biological replicates as n=4 for each figure.
3. Fig. 3B: Does the width of the boxes present the number of bacteria? Yes. Information is shown in the supplemental table. Please indicate this in the main figure to make understanding easier.

Reviewer #3

(Remarks to the Author)

In silico analyses revealed three biosynthetic gene clusters in *P. fluorescens* MYb115. Two were characterized, and the authors demonstrate that a PKS cluster produces sphingolipids, which alters sphingolipid metabolism in the host. An active PKS cluster in MYb115 is required to provide protection against *Bacillus thuringiensis* (Bt), which is (in part) mediated by host sphingolipid metabolism. Thus, a nice interplay between a natural member of the *C. elegans* microbiota, *C. elegans* and microbial pathogens has been uncovered.

Major Comments:

1. Three biosynthetic gene clusters were identified, yet only two were functionally characterized. For completeness, all three BGCs should be included in the analyses for their role in providing protection against Bt247;
2. Are the protective effects specific for Bt247?
3. one cannot conclude that NRPS does not contribute to the protection against Bt247 with an active PKS. The experiments should be repeated in a Δ sga background to address whether Δ sga Δ nrpA reduces survival compared to Δ sga and/or Δ nrpA. The third BGC should also be included.
4. while there is a phenotype observed, what is known about leaky expression of the arabinose promoter in this system?
5. It is expected that gene deletion mutants are complemented for function

Minor Comments:

6. L82: it should read 'intraspecies' and not 'interspecies';
7. L109: this reads odd; please rephrase;
8. Fig. 1: since these are separate infection studies, and not consecutive sampling, data should be shown as bars instead of continuous lines.
9. Fig 5A: please show actual data rather than the abstract interpretation

Version 1:

Reviewer comments:

Reviewer #1

(Remarks to the Author)

This is an interesting, important study. I do not have any further concerns with the manuscript.

Reviewer #2

(Remarks to the Author)

The authors have revised their manuscript with great diligence and have addressed the majority of the concerns raised, particularly those related to the identification of bacterial SLs in the first section. I support the public communication of this work.

However, I am still not fully satisfied with the organization of the figures. The order in which the figures are presented does not always correspond to the sequence in which they are referenced in the Results section. For example, Figure 1C is first mentioned in the second paragraph, while Figures 1D–1F are already discussed in the first paragraph.

Additionally, the main figures present only a limited portion of the results, whereas the supplementary material contains 22 figures, some of which include highly relevant data. Given that many readers may not consult the supplementary information in detail, I strongly recommend that the authors incorporate some of these key findings into the main figures to enhance the accessibility and impact of the manuscript.

Additional minor remark

line 212: two dots after “Figure S4)”

line 279: This may indicate that MYb115-derived SLs do not 280 strongly affect *C. elegans* on the transcript level, but more strongly influence the host on the proteome or metabolome level.

line 305: “we $\Delta\Delta$ integrated” what does $\Delta\Delta$ integrated mean?

Similarly, line 303: colonisation Δ propanoate

Reviewer #3

(Remarks to the Author)

The authors have performed a comprehensive revision of their original submission, and I believe that all of my original concerns have been fully addressed. The (new) data strongly support the authors' conclusions. Their findings present a major advance. The authors must be lauded for their efforts!

Reviewer's Responses to Questions in grey

Our responses in blue

We thank all reviewers for thoughtful reading of the manuscript and the very constructive criticism.

Reviewer #1 (Remarks to the Author):

This is an important study characterizing a polyketide synthase from *Pseudomonas fluorescens* MYb115, a microbe associated with *C. elegans* in nature that has been shown to offer *C. elegans* protection from pathogenic Bt infections. The researchers identified the polyketide synthase (PKS) activity, deleted it, and found that the microbe no longer offers protection against Bt infection in *C. elegans*. They used LC-MS to identify the product of the polyketide synthase, and found that the enzyme produces long-chain sphinganine molecules and then used isotopic labeling to show that serine is incorporated into the sphinganine molecules allowing them to propose a pathway for the activities of the PKS. This is very solid work and is important and interesting because the researchers also identified genes encoding PKS activity in many other microbes, especially those who interact with host organisms. This is a significant finding because the PKS were previously believed to be found in fungi and rarely in bacteria.

The investigators then sought to determine how the PKS products protect *C. elegans* from Bt pathogens. Sphingolipids are important components of the intestinal barrier, so the researchers used transcriptomic and lipidomic analysis to determine changes in the *C. elegans* sphingolipids in worms eating the wild type bacteria vs the bacteria in which the PKS was deleted. Interestingly, they found that the *C. elegans* that were colonized by the wild type MYb115 bacteria had lower levels of many sphingolipid species compared to worms who were colonized by the mutant bacteria lacking PKS. However, this may be minor because only a few species were significantly changed. Researchers then knocked down sphingolipid metabolism genes in *C. elegans* and examined susceptibility to Bt. They found that knocking down enzyme in the de novo pathway leading to ceramide protected worms from infection, while knocking down sphingomyelin synthase, which converts ceramide to sphingomyelin, resulted in more susceptible worms. The MYb115 somewhat rescued this susceptibility of the *sms-1* knockout, although this is puzzling, because the researchers showed that colonization by MYb115 leads to reduced sphingomyelins in worms, so it is unclear how this rescue occurred.

Our reply: We thank the reviewer for this valid comment. We agree that we do not understand how exactly MYb115 interacts with *C. elegans* sphingolipid metabolism. Sphingolipid homeostasis is controlled by a complex network comprising several levels of regulation. The enzymes responsible for sphingolipid production and turnover comprise a metabolic network that gives rise to numerous bioactive molecules, which participate in highly complex and interconnected pathways influencing a multitude of physiological processes (Hannun and Obeid, 2018 <https://doi.org/10.1038/nrm.2017.107>). Also, sphingolipid metabolism shares common substrates with other metabolic routes and is, for example, highly connected to other lipid metabolic networks. Consequently, imbalances in sphingolipid metabolism in a mutant may have far-reaching consequences for host physiology. Thus, the effect of MYb115 on the phenotype of a sphingolipid metabolism

mutant (increased survival after Bt infection) may not be directly linked to its effect on wildtype worms (decrease in sphingomyelin).

Nevertheless, our comprehensive data sets, including several new data sets now added to the revised manuscript, do allow us to draw several important conclusions on commensal-mediated immune protection. Important new insights are for example that sphingolipid-producing MYb115 does cause a decrease of certain host sphingolipid species, including sphingomyelin species, in comparison to non-sphingolipid-producing MYb115. Moreover, MYb115 ameliorates the survival phenotype of *C. elegans* sphingolipid enzyme mutants following Bt infection in comparison to OP50. We now integrated a critical evaluation of our data into the discussion and further clarify that the current comprehensive data sets do not yet provide a final answer on whether the observed effect of MYb115 on the survival phenotype of the sphingolipid enzyme mutant is direct (on sphingolipid metabolism) or indirect (p. 17 and 18, line 495-501).

Also, we have now invested considerable efforts in further improving our understanding of MYb115 sphingolipid biosynthesis and the effects produced by MYb115, including biochemical characterization of the enzymes encoded in the biosynthetic gene cluster (BGC) that catalyze MYb115 sphingolipid biosynthesis (in collaboration with experts in sphingolipid biosynthesis), further metabolic analyses of bacterial cultures (in collaboration with experts in metabolic analyses), and further phenotypic analyses in *C. elegans* demonstrating – among others – that protection against another Bt strain, Bt679, and protection of the *C. elegans* intestinal barrier following Bt infection depends on MYb115-produced sphingolipids and that MYb115-mediated protection is independent of two known *C. elegans* Bt defense pathways (p38 and JNK MAPK pathways), the mitochondrial surveillance response, and of a Bt toxin glycosphingolipid receptor (please also see our answers to the reviewers' comments below).

This paper is well written and the data are clearly and thoroughly presented. While the *C. elegans* experiments did not clarify how the sphinganine products from MYb115 lead to protection from Bt infection in *C. elegans*, the observations are important starting points for future experiments. The researchers used rather vague terms, for example on line 309 “These data indicate that MYb115 interacts with host sphingolipid metabolism.....” While the nature of the interaction isn't clear, at this point I think this is a valid statement, because more definitive mechanism can't be determined from the data.

Our reply: We thank the reviewer for their comments.

One technical comment. When researchers receive deletion lines from the CGC (ok alleles) or NBRP (tm alleles) they typically perform outcrosses against wild type worms since the original deletion strains can contain many other mutations. The researchers should report how many times the strains were outcrossed prior to their experiments, or else say that they were not outcrossed.

Our reply: We genotyped the respective sphingolipid metabolism pathway mutant and thus confirmed the deletion alleles, but we did not outcross the mutants. We now clarify that the mutants were not outcrossed in the materials and methods section and show the detailed genotyping results in a supplementary figure (Figure S26).

Reviewer #2 (Remarks to the Author):

The manuscript Peters et al. identified an iterative type I polyketide synthase (PKS) in MYb115 and showed that PKS regulates bacterial sphingolipid biogenesis and this interferes with sphingolipid metabolism of the host, thus exerting a protective role for the host against pathogen infection. The paper contains a number of interesting observations, especially identification of an SPT-independent mechanism for sphingolipid biogenesis.

Our reply: We agree with the reviewer that the discovery of a new way of bacterial sphingolipid synthesis significantly advances our knowledge on bacterial sphingolipid metabolism. While the few known bacterial sphingolipid producers (such as *Bacteroidetes*), like eukaryotes, produce sphingolipid as primary metabolites in a manner that depends on the enzyme serine palmitoyltransferase (SPT), *P. fluorescens* MYb115 produces sphingolipids as secondary metabolites in a non-canonical way that depends on a BGC, the polyketide synthase (PKS) *PfSgaB*. To our knowledge this is the first example of a bacterial PKS shown to be involved in sphingolipid biosynthesis and also the first description of a *Pseudomonas* isolate as sphingolipid producer.

Moreover, in the revised manuscript we now present and discuss a comprehensive set of new results that considerably expand our knowledge on the *P. fluorescens* BGC-dependent sphingolipid biosynthesis pathway: We previously could only assume that the enzyme encoded by *PfSgaB* within the BGC substitutes the function of SPT - the above-mentioned rate-limiting enzyme for sphingolipid biosynthesis in all eukaryotes and the few bacterial sphingolipid producers. In the revised manuscript, we now integrated comprehensive new data, which we generated in collaboration with Dominic Campopiano, Michael Herrera, and Francesca Lubbock (University of Edinburgh, all now co-authors) on the function of *PfSgaB*. Using heterologous expression in *E. coli* and subsequent *in vitro* functional analysis with the purified enzyme we could prove that *PfSgaB* functions as SPT. Furthermore, we identified a putative short chain dehydrogenase/reductase (SDR) in the MYb115 sphingolipid BGC, which is predicted to share structural homology with 3-ketodihydrospinganine reductase (KDSR). KDSR is known to catalyze the reduction of 3-KDS to dihydrospinganine (DHS) (Beeler et al., 1998. <https://doi.org/10.1074/jbc.273.46.30688>; Fornarotto et al., 2006. <https://doi.org/10.1016/j.bbalip.2005.11.013>); whilst this step is ubiquitous in eukaryotic sphingolipid biosynthesis, it is unusual in bacterial sphingolipid pathways (Stankeviciute et al., 2022. <https://doi.org/10.1038/s41589-021-00948-7>). The inclusion of this eukaryotic-like step further distinguishes the *P. fluorescens* sphingolipid-producing BGC from canonical bacterial sphingolipid biosynthesis. In the revised manuscript, we now highlight these new exciting findings in a new section and new figure (see lines 204-241 and Figure 2).

However, the data did not provide a significant novelty and new insights into mechanistic explanation for microbiota-host interaction in regulation of host immune response, given that the role of sphingolipid in immune response and the interplay between bacterial sphingolipid and host sphingolipid biogenesis/metabolism are known.

Our reply: Many thanks for this comment. We politely disagree. Our work very clearly and substantially advances our knowledge on the functional significance of a bacterial, sphingolipid-producing PKS in microbiota-mediated protection against pathogens. We found this PKS in many other bacteria that, like *Pseudomonas*, are not yet known sphingolipid producers. This is a significant novelty since the interplay between bacterial sphingolipids and host sphingolipid metabolism has so far only been reported for one bacterial phylum and one host, the *Bacteroidetes* in the human gut (e.g. Johnsen et al., 2020. <https://doi.org/10.1038/s41467-020-16274-w>; Le et al., 2022. <https://doi.org/10.1016/j.chom.2022.05.002>; Brown et al., 2019. <https://doi.org/10.1016/j.chom.2019.04.002>). Most importantly, to date, the relevance of this interplay has never been reported in the context of microbiota-mediated protection against pathogens. Also, we show that a *Pseudomonas* species alters sphingolipid metabolism in *C. elegans* and establish the importance of *C. elegans* sphingolipid metabolism for survival after Bt infection. We agree with the reviewer that a more detailed study on the mechanism by which MYb115-derived sphingolipids modify host sphingolipid metabolism and/or immune response is an interesting topic for a future study. However, such an analysis is a manuscript of its own and thus goes beyond the scope of the present study. In response to the reviewer's comment, we now more clearly highlight the importance of our numerous novel insights and further point to promising future research directions.

In addition, some of the claims are also not sufficiently supported by the results.

Our reply: We hope that we could sufficiently address this concern (see our answers below).

For these reasons, I am not recommending publication in Nature Communications.

Major issues:

1. *sgaAB* encoded PKS catalyses sphingolipid biosynthesis and improves resistance of animals against Bt247.

Our reply: We thank the reviewer for bringing to our attention the ambiguities in MYb115-mediated protection listed below. In the revised manuscript, we have added a new paragraph (line 264-299) summarizing the new insights we have gained on how the host response to Bt247 is affected by MYb115 and MYb115 Δ *sgaA*.

a) Is the protective effect of MYb115 linked with altered activity of known pathways for worm innate immunity, e.g. p38, MAPK, JUN-1, ELT-2, necrosis pathway or *bre* genes?

Our reply: Following the reviewer's suggestion, we tested the involvement of the p38 MAPK and the JNK-like MAPK pathway in MYb115-mediated protection against Bt247. We found that the protective effect is completely independent of the p38 MAPK pathway (MYb115 also protects the p38 MAPK pathway *tir-1*, *nsy-1*, *sek-1*, and *pmk-1* mutants) or the JNK-like MAPK KGB-1 (Figure S19). We could previously already exclude an involvement of the *bre* genes in *C. elegans* defense against Bt247, given that *bre* mutants are susceptible to Bt247 infection (see discussion line 511-518). However, because of the direct link between *bre* genes and the biogenesis of complex glycosphingolipids, we confirmed these previous findings and included the data in Figure S25B. Please also see our response to the reviewer's comment 3b below.

b) How does MYb115 affect toxicity of the other Gram-positive pathogen, e.g. Bt679?

Our reply: We have previously already shown that MYb115 also protects against another Bt strain, Bt679 (Kissoyan and Peters et al. <https://doi.org/10.3389/fcimb.2022.775728>). We now mention this in the revised manuscript (line 272-274). The comments by reviewer #2 and #3 prompted us to test if protection against Bt679 is also lost on the MYb115 Δ *sgaA* mutant and indeed it is. We now included this new additional result in Figure S17. We did not test other Gram-positive bacteria.

c) Can the MYb115 Δ *sgaAB* phenotype be rescued by restoring *sgaAB* expression in this mutant bacterium?

Our reply: In response to the comments of reviewer #2 and #3 we have now conducted a series of experiments to functionally complement the MYb115 Δ *sgaA* mutant by inserting the vanillic acid inducible PvanCC promoter in front of *sgaA* or the complete *sgaAB* BGC on the plasmid pSEVA631, which was then introduced into the MYb115 mutant. Only the complementation that included the complete *sgaAB* BGC restored sphingolipid production (detection of compounds 1 (m/z 414.4 [M+H]⁺), 2 (m/z 386.4 [M+H]⁺), and 3 (m/z 442.4 [M+H]⁺) by LC-MS (Figure S27). We observed a clear increase in resistance to Bt247 of worms on this complemented MYb115 Δ *sgaA* mutant and now present these additional new results in Figure 1F of the revised manuscript.

Of note: We obtained two complemented MYb115 Δ *sgaA* mutant strains. While these strains were meant for targeted activation of the BGC, we realized that the vanillic acid inducible promoter was leaky (and more so in one strain than in the other), resulting in sphingolipid production also in the absence of vanillic acid. However, we used this to show that variations in sphingolipid production are reflected in variations in the protective effect, providing further evidence that host protection is dependent on bacterial sphingolipid production. The whole set of obtained results do clearly demonstrate the role of the complete *PfSgaAB* BGC in sphingolipid production and host protection, which we now emphasize in the revised manuscript (line 179-192 and Figure 1H).

d) Can expression of *sgaAB* in another bacterium which normally doesn't have a protective role against Bt247 result a protection?

Our reply: This is an interesting point. Yet, it is not essential to demonstrate a causal effect of *sgaAB* on protection against pathogenic Bt. Such evidence was produced with MYb115, which we now substantiated with the complementation of the MYb115 Δ *sgaA* mutant. Therefore, we decided to prioritize these additional complementation experiments (requested by reviewer #2 and #3), in order to further advance our understanding, and as a consequence, we did not include the here proposed experiment at this time.

e) How is the intestine morphology and integrity upon Bt247 infection affected by MYb115 and *sgaAB*?

Our reply: MYb115 limits Bt-induced damage to the intestinal epithelium, as we already demonstrated previously (Kissoyan et al., 2019. <https://doi.org/10.1016/j.cub.2019.01.050>). Prompted by the reviewer's question, we now further tested if MYb115-derived sphingolipids are involved in mitigating Bt-induced damage. Using the *C. elegans* PGP-1::GFP reporter strain, we confirmed that MYb115 PKS-derived sphingolipids reduced membrane damage caused by Bt infection, while the MYb115 Δ *sgaA* mutant did not. We included these new results in the revised manuscript (line 290-299. Figure S20).

f) Line 146: MYb115 lipidomic analysis was performed to test whether the bacteria sphingamines exist as free compounds or are part of lipids and sphingamines 1-3 and PG-sphingolipids 4-6 were identified. Does this result suggest that the bacteria sphingamines exist as both free compounds and part of lipids? A clear conclusion to the question should be provided to make understanding easier, especially for those who are not familiar with sphingolipid metabolism. In addition, I don't understand why elucidating the existing form of sphingamines is important to know how MYb115 interact with the host. Additional background explanations are required to understand the underlying logic. The author concluded that they are not able to differentiate between the effects of the individual sphingolipid species. I don't think that lipidomic analysis alone would help to answer this question. But supplementation of these sphingamines or PG-sphingolipids might do.

Our reply: Many thanks for these valuable comments. The MYb115 lipidomic analysis using high-resolution Liquid Chromatography Tandem Mass Spectrometry (HRES-LC-MS/MS) allowed us to identify the PG-sphingolipids 4-6. We now clarify this (line 167-170). The supplementation experiments would require purified sphingolipids from MYb115, which however is not available and thus, could not easily be obtained. Therefore, we decided for a different approach to further elucidate the involvement of different sphingolipids. In collaboration with Manuel Liebeke (Kiel University, now co-author) using MALDI mass spectrometry spot assays (<https://doi.org/10.1038/s41596-023-00864-1>), we visualized sphingamines 1-3 and PG-sphingolipid 4-6 in bacterial cultures, whose protective effect we then tested in survival analyses. These additional experiments demonstrate that protection significantly correlates with the abundance of sphingamines 1-3 and PG-sphingolipid 4, indicating that host protection is dependent on these sphingolipids. In the revised manuscript, we now added and explain these new data (line 179-192 and Figure 1H).

2. Transcriptome study:

a) it is surprising that Bt247 infection only cause few numbers of DEG in MYb115 and worms on the MYb115 Δ *sgaAB* mutants, given their significantly different survival rate. As the RNA-seq results was not shown in any main figures or supplemental data, expect an enrichment analysis in Fig. 4A and S4A, it is difficult to make much comments.

Our reply: The transcriptome response of *C. elegans* to MYb115 and MYb115 Δ *sgaA* is indeed very similar under both conditions, infected and non-infected. We agree with reviewer #2 that this result is surprising and unexpected. As discussed also below, the only difference between MYb115 and MYb115 Δ *sgaA* is the production of sphingolipids. We do think that sphingolipid production may not strongly affect *C. elegans* on the transcript level, but more strongly influences the host on the proteome/metabolome level. We mention this in the

revised manuscript in line 279-281. We thank the reviewer for pointing out that we did not clearly present the RNAseq data in the manuscript. We now added a short description of the results in the context of host pathogen defense (line 274-287) and present the data in Figure S18 and Table S7. As before, raw data and processed data are accessible through GEO Series accession number GSE245296 at NCBI's Gene Expression Omnibus.

Does Bt247 infection affect transcriptome significantly? Does it result in activation of innate immunity genes?

Our reply: Yes, we have previously shown in several independent studies that Bt247 infection has a strong effect on the *C. elegans* transcriptome and results in activation of *C. elegans* pathogen-responsive/innate immunity genes (Boehnisch et al., 2011.

<https://doi.org/10.1371/journal.pone.0024619>. Nakad et al., 2016.

<https://doi.org/10.1186/s12864-016-2603-8>. Yang, Dierking et al., 2015.

<https://doi.org/10.1016/j.dci.2015.02.010> Zarate-Potes et al., 2020.

<https://doi.org/10.1371/journal.ppat.100882>).

If Bt247 does not cause a significant alteration in gene expression, it would be hard to find involved factors via comparing transcriptome of animals on MYb115 and on the MYb115 Δ sgaAB mutants. In a such scenario, MYb115 might protect animals via other mechanisms that are independent of influencing transcript level.

How does MYb115 affect transcriptome without Bt247 infection?

Our reply: For this study, we specifically compared the *C. elegans* transcriptome response to MYb115 in comparison to the non-sphingolipid producing MYb115 Δ sgaA mutant. We did not include the laboratory food bacterium *E. coli* OP50 as control, which would be necessary to assess the general effect of MYb115 on the *C. elegans* transcriptome without Bt247 infection. The comparison between the *C. elegans* response to OP50 and MYb115 was not the subject of the study.

If the Bt247 infectious bacteria remodel the worms' transcriptome much more than MYb115 does, the the potential changed genes that are only influenced by MYb115 or MYb115 Δ sgaAB mutants might submerge among Bt247-caused alteration in gene expression.

Our reply: We thank the reviewer for their comment. This may indeed be the case and, as discussed above, MYb115-derived sphingolipid may affect the host response more strongly on the proteome level. We now address this aspect in the revised manuscript (line 279-281).

b) The transcriptome data was then integrated into the iCEL1314 genome-scale metabolic model and 24 and 23 significant differences in the presence or absence of Bt247 were received. What does "24 and 23 significant differences" mean? Does it refer to Differences between generated metabolic models? The results must be described in more details so that non-experts in iCEL1314 could understand the data.

Our reply: In the methods section of the revised manuscript we did explain more in detail what the different data types are and how metabolic models were applied. In the revised results, we now point to the methods section for these details (line 307).

Additionally, we have now simplified the data types that we analysed. Previously, we had the OFD results, lower bound and upper bound – we performed statistical analysis on them separately. However, the interpretation of differences was difficult. In the revised manuscript, we now combined the lower and upper bound values into one score – the center, which represents: $(\text{upper bound} - \text{lower bound}) / 2$. Centers are more representative of the solution spaces for each of the reactions, and coefficients are easier to interpret. After doing so, a few of the previous reactions are no longer present among the significant results – instead of 24 (KO vs WT with Bt247) and 23 (KO vs WT without Bt247), we obtained 16 reactions in each contrast. We have updated the manuscript accordingly (line 305-322).

What does the "Gene Ratio" in the Figure 4A mean? Fig. 4A and S4A only show enrichment of metabolic processes even without notifying whether these processes are activated or

inactivated by *sgaAB* in MYb115 (whether genes involved are up- or down-regulated). A list of genes used for the enrichment analysis would also be important to understand the result.

Our reply: We would like to thank the reviewer for drawing our attention to this point. We did not use genes for enrichment, rather the reactions significant after our statistical analyses. The “pathway” universe were the subsystem annotations of all reactions within the model. Since we identified only few reactions as significantly up- or down-regulated an enrichment on them separately would not yield any results. We changed the mislabeled “Gene ratio” to “Reactions”. The individual significant reactions are provided in Table S10.

Importantly, our goal for this analysis was to understand which pathways are affected, not necessarily specify which reactions are affected. Additionally, we do not want to show the coefficients in the main manuscript because they can be misinterpreted in the traditional sense due to the fact that in metabolic models, the positive or negative sign of reactions actually represents the directionality of a reaction. For example, a reaction can be irreversible, e.g. substrate -> product, or reversible, i.e. substrate <-> product. In our hypothetical irreversible reaction, the reaction flux can only have positive values (it's irreversible), so here a negative coefficient would mean that the flux through this reaction in the worms grown on mutant bacteria is lower. However, for our hypothetical reversible reactions, values can be both positive and negative. So, if all our values were negative, a positive coefficient would actually mean that the worms grown on mutant bacteria would have less flux (e.g. -0.5 vs -2) through this reaction. Since we have a large mix of reversible and irreversible reactions within our network, and we have not only looked at flux potentials, but also at mathematical optimal flux distribution (which is the optimal but not unique solution to the linear optimization problem), we cannot reliably infer the significant involvement of individual reactions based on these statistical analysis results. Nevertheless, we can see trends from our analysis on the pathway level, which is why we have described our results as we have. These aspects have now been briefly explained in the methods section (line 701-725).

c) Line 220: “Here, we also saw an enrichment in valine, leucine, and isoleucine degradation, which are branched-chain amino acids (BCAA). This pathway is directly connected with propanoate metabolism that provides components for the synthesis of the C15iso fatty acid, which is the precursor for sphingolipids in *C. elegans*”: are these genes up- or down-regulated in MYb115 vs. worms on the MYb115 Δ *sgaAB* mutants? Again, provide more RNA-seq information will avoid such a confusion.

Our reply: In Table S10 we have added the information about the genes that encode the significant reactions, as well as the logFC values of these genes from the transcriptomic analysis. We would like to point out that the iMAT++ algorithm used to integrate the transcriptomic data into the iCEL1314 metabolic model is done on a sample basis, therefore any statistical comparisons of gene expression is not taken into account during this process. This means that comparing the differences in simulation results of reactions encoded by a certain gene and the logFC values of this gene might not directly match in direction. This is a desirable attribute of metabolic modelling since we can predict metabolic requirements of an organism that are in conflict with the gene expression differences – these might be caused by post-translational modifications or other effects. These aspects have now been briefly explained in the methods section (line 710-716).

d) The RNA-seq data should be validated via qRT-PCR.

Our reply: The metabolic network analysis, which is based on the RNAseq data, revealed that sphingolipid metabolism reactions show differential activity between MYb115 and MYb115 Δ *sgaA*. We focused on this result and validated the transcriptomic data using lipidomic profiling. The other results of the RNAseq study are not the focus of this project.

3. Host sphingolipid metabolism interferes with Bt247 susceptibility

a) From the host lipidomics analysis shown in Fig. 4b, *sgaAB* in MYb115 seems to reduce

level of Cer, HexCer, DhCer and SM in the host. To test whether accumulation of these molecules reduces survival of animals upon Bt247 infection, the authors used different mutants in the sphingolipid metabolism pathway to test their susceptibility to Bt247. However, *cgt-1* and *cerk-1* mutant animals fed with OP50 show increased survival, although these animals are supposed to have Cer accumulation. In contrast, *asm-3*, *hyl-1*, *hyl-2*, *sptl-1* and *sptl-3* mutants which probably have reduced Cer level are also resistant. Do these results argue against an important role of Cer in Bt247 susceptibility? In summary, from these Bt247 survival assay with different mutants in sphingolipid metabolism pathway, the author could not provide a clear answer which classes of host sphingolipid are responsible for the altered susceptibility to the pathogen.

Our reply: We thank the reviewer for this valid comment. We tested four ceramide metabolic gene mutants, all with an expected increased ceramide content, but two mutants (*cerk-1* and *cgt-1*) show a variable survival phenotype and two mutants (*asah-1,2* and *sms-1*) are more susceptible to infection. In response to the comments of reviewer #2 we now further assessed the role of specific sphingolipids in defense against Bt infection and supplemented Bt infected worms, using a set of new experiments with the commercially available sphingolipids ceramide, sphingomyelin, and sphingosine-1-phosphate. We found that supplementation with C18 and C20 ceramide significantly improved survival rates, while C22 ceramide, C16 sphingomyelin, C18 sphingomyelin, d-sphingosine, and S1P did not affect survival. These new results and the phenotypic differences between the ceramide metabolic gene mutants imply that the inhibition of ceramide metabolism (and the associated increase in ceramide content) is not the only factor determining susceptibility to infection. We present and discuss these additional new results in the revised manuscript (line 390-396. Figure S13 and line 483-492).

b) What is account for Bt247 toxicity? Why should altered sphingolipid metabolism influence toxicity of Bt247?

Our reply: How altered host sphingolipid metabolism affect defense against Bt and thus Bt toxicity remains mysterious. Since sphingolipids are required for the integrity of cellular membranes, but can also act as bioactive signalling molecules involved in regulation of a myriad of cell activities, there also are a myriad of potential links between sphingolipid metabolism and defense against Bt247. In response to the reviewer's comments above, but also below, we further explored the two most apparent potential links between sphingolipids and *C. elegans* defense against Bt infection: The involvement of mitochondrial surveillance (see comment by the reviewer below) and the *bre* genes (see comment 1 a). We did not find any evidence of a link between Bt infection and mitochondrial surveillance or the *bre* genes. We present and discuss these results in the revised manuscript (line 426-438. Figure S25 and line 504-518). Based on our comprehensive data set, we discuss possible alternative routes of how sphingolipid metabolism interacts with Bt247 (line 502-522).

A study from Ruvkun's lab (Liu et al., 2014, Nature) has shown that some nature habitat bacteria cause mitochondria dysfunction and animals respond with mitochondrial surveillance machinery which is ceramide dependent. At the same time some other bacteria could inhibit mitochondrial surveillance to render a more effective virulence. Does Bt247 modulate the mitochondria surveillance?

Our reply: Following the reviewer's suggestion, we tested if Bt247 infection induces the mitochondrial stress-induced *hsp-6p::gfp* reporter that was also used in the study by Liu et al. Neither Bt247 infection, nor MYb115 induced expression of *hsp-6p::gfp*, indicating that mitochondrial surveillance is not involved in MYb115-mediated protection against Bt247 infection. We added these results to the revised version of our manuscript (line 426-438. Figure S25A).

4. Sphingolipid in MYb115 affect host sphingolipid biogenesis and metabolism.

There is no attempt to address how this could happen. Is uptake of the bacterial sphingolipid into worm intestine necessary? Can feeding worms with the identified sphinganine 1-3 or sphingolipid 4-6 impact Bt247 resistance? If yes, does this supplementation change sphingolipid composition of worms?

Our reply: We agree with the reviewer that the question how exactly MYb115-derived sphingolipids affects host sphingolipid metabolism is highly intriguing, yet its further analysis goes beyond the scope of the current study. Our study did reveal for the first time that *P. fluorescens*-derived sphingolipids affect *C. elegans* sphingolipid metabolism and that modulation of host sphingolipid metabolism increases tolerance towards Bt infection – among others. We now provide a comprehensive set of new data to substantiate the findings made. Supplementation experiments, as suggested by the reviewer, would indeed be helpful to differentiate between the effects of the identified sphinganine **1-3** and PG-sphingolipids **4-6**. As also discussed above (comment 1 f), we unfortunately do not have the MYb115-derived sphingolipids purified, which we would need to do these supplementation experiments. However, in collaboration with Manuel Liebeke (Kiel University, now co-author) using MALDI mass spectrometry spot assays (<https://doi.org/10.1038/s41596-023-00864-1>), we visualized sphinganine **1-3** and PG-sphingolipid **4-6** in bacterial cultures, whose protective effect we then tested in survival analyses. These additional experiments demonstrate that protection significantly correlates with the abundance of sphinganine **1-3** and PG-sphingolipid **4**, indicating that host protection is dependent on these sphingolipids. In the revised manuscript, we now added and explain these new data (line 179-192 and Figure 1H).

Minor issues

1. Zeile 377: Notably, in *C. elegans*, glucosylceramide deficiency was linked to an increase in autophagy which plays an important role in cellular defence after attack by certain Bt pore-forming toxins (PFTs). Does Myb affect autophagy? If yes, could manipulating autophagy affect survival of animals to Bt247?

Our reply: We appreciate the reviewer's suggestion regarding the possible involvement of autophagy. This is indeed an interesting idea and could provide valuable insights for future work. However, due to time and resource constraints, we prioritized addressing the most important points in our revisions. Thus, we decided against an additional investigation of autophagy.

Fig. S6: Should A1 be compared with B1, A2 with B2 and so on? If yes, why is the mutant A10 in red line and B10 in yellow? The color used for *asah-1* and *asah-2* are too similar to be differentiated. Maybe just add the name of the mutant directly to the figures. Why are there several figures for some mutants, e.g. *asah-1* while for some others only one e.g. *cgt-1* on OP50? It does not seem to be biological replicates as n=4 for each figure.

Our reply: We appreciate the reviewer's feedback regarding the data representation. In response, we have revised the figure to improve clarity. Specifically, we have updated the color scheme: for all assays where worms were exposed to OP50, the lines are now depicted in grey (panel A), while survival assays with worms exposed to MYb115 are shown in blue (panel B). This adjustment ensures consistency with the color scheme used throughout the manuscript. Additionally, as suggested, we have directly labeled each facet with the name of the corresponding mutant strain, further enhancing the figure's clarity. Moreover, we have now added more survival assays, yielding a larger number of independent experiments for each *C. elegans* mutant strain, and thus much more robust insights into the role of the respective genes. For some mutants i.e. (*cgt-1* and *cerk-1*) the additional experiments revealed that the survival phenotype is variable. We thus included the technical and biological replicates in the heat-map of Figure 5C, so that the reader can see

at a glance how many experimental runs were done and identify variation across technical and biological replicates. The previous and the additional new data now provide an in-depth as well as statistically sound overview of the involvement of different branches of the entire sphingolipid metabolism.

3. Fig. 3B: Does the width of the boxes present the number of bacteria? Yes. Information is shown in the supplemental table. Please indicate this in the main figure to make understanding easier.

Our reply: We thank the reviewer for pointing out this ambiguity. We added the information in the legend of Figure 3.

Reviewer #3 (Remarks to the Author):

In silico analyses revealed three biosynthetic gene clusters in *P. fluorescens* MYb115. Two were characterized, and the authors demonstrate that a PKS cluster produces sphingolipids, which alters sphingolipid metabolism in the host. An active PKS cluster in MYb115 is required to provide protection against *Bacillus thuringiensis* (Bt), which is (in part) mediated by host sphingolipid metabolism. Thus, a nice interplay between a natural member of the *C. elegans* microbiota, *C. elegans* and microbial pathogens has been uncovered.

Major Comments:

1. Three biosynthetic gene clusters were identified, yet only two were functionally characterized. For completeness, all three BGCs should be included in the analyses for their role in providing protection against Bt247;

Our reply: We thank the reviewer for their comments. Although interesting, analysis of the third BSG is not essential to prove involvement of the *sgaAB* BGC in sphingolipid metabolism and protection against pathogens. We decided to prioritize on other experiments which now yielded a comprehensive data base for obtaining critical additional insights into the role of *sgaAB* BGC (see also above and below replies), and thus omitted an analysis of the arylpolyene pathway.

2. Are the protective effects specific for Bt247?

Our reply: We have previously already shown that MYb115 also protects against another Bt strain, Bt679 (Kissoyan and Peters et al. <https://doi.org/10.3389/fcimb.2022.775728>). We now mention this in the revised manuscript (line 272-274). The comments by reviewer #2 and #3 prompted us to test if protection against Bt679 is also lost on the MYb115 Δ *sgaA* mutant and indeed it is. We now included this result in Figure S17. We did not test other Gram-positive bacteria.

3. one cannot conclude that NRPS does not contribute to the protection against Bt247 with an active PKS. The experiments should be repeated in a Δ *sga* background to address whether Δ *sga* Δ *nrpA* reduces survival compared to Δ *sga* and/or Δ *nrpA*. The third BGC should also be included.

Our reply: We thank the reviewer for pointing out this ambiguity. Due to time and resource constraints, we prioritized addressing the complementation of the MYb115 Δ *sgaA* and Δ *sgaB* mutants requested by reviewer #2 and #3. We now refrain from drawing conclusions regarding the NRPS cluster and rephrased the sentence in line 117-120 “While the PKS gene cluster affects MYb115-mediated protection, we did not observe significant differences in worm survival with or without arabinose supplementation on the MYb115 $P_{BADnrpA}$ strain (Figure 1B, Table S1), indicating that the MYb115 NRPS gene cluster is not involved in MYb115-mediated protection against Bt infection.” to “While the PKS gene cluster affects MYb115-mediated protection, we did not observe significant differences in worm survival with or without arabinose supplementation on the MYb115 $P_{BADnrpA}$ strain (Figure 1B, Table S1). We therefore focused on the MYb115 PKS gene cluster in our subsequent analyses.”

Since we observed a complete loss of sphingolipid production and a clear decrease in host protection for the MYb115 Δ sgaA and MYb115 Δ sgaB single mutants, we think that it is justified to focus on the PKS cluster within the scope of the current project.

4. while there is a phenotype observed, what is known about leaky expression of the arabinose promoter in this system?

Our reply: We assessed MYb115 P_{BAD} sga sphingolipid production in an induced (+ arabinose) and non-induced (- arabinose) state by LC-MS and did not detect any sphingolipids in the non-induced state (Figure 1C). In addition, in the revised manuscript, we included MYb115 P_{BAD} sga in an induced (+ arabinose) and non-induced (- arabinose) state in an experiment to visualize production of the different sphingolipids (collaboration with Manuel Liebeke, Kiel University now co-author) and could confirm that P_{BAD} is not leaky in this system (datasets: MPIMM_514_QE_P https://metaspace2020.org/dataset/2025-02-27_13h37m58s).

5. It is expected that gene deletion mutants are complemented for function

Our reply (also see our response to the comment 1c of reviewer #2 above): In response to the comments of reviewer #2 and #3 we have conducted a series of additional new experiments to functionally complement the MYb115 Δ sgaA mutant by inserting the vanillic acid inducible PvanCC promoter in front of sgaA or the complete sgaAB BGC on the plasmid pSEVA631, which was then introduced into the MYb115 mutants. Only the complementation that included the complete sgaAB BGC restored sphingolipid production (detection of compounds 1 (m/z 414.4 [M+H]⁺), 2 (m/z 386.4 [M+H]⁺), and 3 (m/z 442.4 [M+H]⁺) by LC-MS (Figure S27) and only in the MYb115 Δ sgaA mutant. We observed a clear increase in resistance to Bt247 of worms on this complemented MYb115 Δ sgaA mutant and present the results in Figure 1F of the revised manuscript.

Of note: We obtained two complemented MYb115 Δ sgaA mutant strains. While these strains were meant for targeted activation of the BGC, we realized that the vanillic acid inducible promoter was leaky, resulting in sphingolipid production also in the absence of vanillic acid (datasets: MPIMM_514_QE_P https://metaspace2020.org/dataset/2025-02-27_13h37m58s). However, we used this to show that variations in sphingolipid production are reflected in variations in the protective effect, providing further evidence that host protection is dependent on bacterial sphingolipid production. The whole set of obtained results do clearly demonstrate the role of the complete MYb115 sgaAB BGC in sphingolipid production and host protection, which we now emphasize in the revised manuscript (line 179-192 and Figure 1H).

Minor Comments:

6. L82: it should read 'intraspecies' and not 'interspecies';

Our reply: We have now replaced "interspecies" with "intraspecies"

7. L109: this reads odd; please rephrase;

Our reply: The sentence: "Supplementation of the *C. elegans* laboratory food *Escherichia coli* OP50 with arabinose did not affect survival, showing that arabinose itself does not influence *C. elegans* resistance to Bt (Table S1)." has now been reworded to: "Arabinose supplementation had no effect on resistance of *C. elegans* to Bt infection on its standard laboratory food *Escherichia coli* OP50."

8. Fig. 1: since these are separate infection studies, and not consecutive sampling, data should be shown as bars instead of continuous lines.

Our reply: The continuous lines in Figure 1 A, B and E represent a dose-response relationship. As it is common practice for dose-response curves, the data points are connected.

9. Fig 5A: please show actual data rather than the abstract interpretation

Our reply: We show the survival data in Figure 5C as heatmaps to summarize the results of 32 and 24 survival assays on *E. coli* OP50 and *P. fluorescens* MYb115, respectively, and to facilitate the comparison of results between different knockout mutants. We now also

included technical and biological replicates, so that the reader can see at a glance how many experimental runs were done and to facilitate the identification of variation across technical and biological replicates. For the interested reader, we do provide each individual survival curve in the supplementary Figure S23.

Reviewer's Remarks to the author in grey

Our responses in blue

We thank all reviewers for thoughtful re-evaluation of the manuscript and their comments.

Reviewer #1 (Remarks to the Author):

This is an interesting, important study. I do not have any further concerns with the manuscript.

Our reply: We thank the reviewer for their comments.

Reviewer #2 (Remarks to the Author):

The authors have revised their manuscript with great diligence and have addressed the majority of the concerns raised, particularly those related to the identification of bacterial SLs in the first section. I support the public communication of this work.

However, I am still not fully satisfied with the organization of the figures. The order in which the figures are presented does not always correspond to the sequence in which they are referenced in the Results section. For example, Figure 1C is first mentioned in the second paragraph, while Figures 1D–1F are already discussed in the first paragraph.

Our reply: We changed the text according to the reviewer's suggestions.

Additionally, the main figures present only a limited portion of the results, whereas the supplementary material contains 22 figures, some of which include highly relevant data. Given that many readers may not consult the supplementary information in detail, I strongly recommend that the authors incorporate some of these key findings into the main figures to enhance the accessibility and impact of the manuscript.

Our reply: We thank the reviewer for pointing this out. We now incorporated the data shown in figures S13A into figure 2D, figure S18 B is now (new) figure 4A, figure S19 is now incorporated in figure 4 (new figure 4B and C), and figure S20 is now (new) figure 5. In addition, we combined several supplementary figures, so that we now have a total of 21 (instead of 27) supplementary figures.

Additional minor remark

line 212: two dots after "Figure S4)"

Our reply: We removed one dot.

line 279: This may indicate that MYb115-derived SLs do not strongly affect *C. elegans* on the transcript level, but more strongly influence the host on the proteome or metabolome level.

Our reply: We changed the sentence according to the reviewer's suggestion.

line 305: "we $\Delta\Delta$ integrated" what does $\Delta\Delta$ integrated mean?

Our reply: We thank the reviewer for pointing out this mistake. We removed the ' $\Delta\Delta$ '.

Similarly, line 303: colonisation Δ propanoate

Our reply: We removed the 'colonisation Δ '.

Reviewer #3 (Remarks to the Author):

The authors have performed a comprehensive revision of their original submission, and I believe that all of my original concerns have been fully addressed. The (new) data strongly

support the authors' conclusions. Their findings present a major advance. The authors must be lauded for their efforts!

Our reply: We thank the reviewer for their encouraging comments.